# Do Object Channels Improve Robustness in Deep Reinforcement Learning?

**Jannis Blüml**\*
*The Hessian Center for Artificial Intelligence (hessian.AI)*
*Technische Universität Darmstadt*

*jannis.blueml@tu-darmstadt.de*

**Cedric Derstroff**\*
*The Hessian Center for Artificial Intelligence (hessian.AI)*
*Technische Universität Darmstadt*

*cedric.derstroff@tu-darmstadt.de*

**Bjarne Gregori**
*Technische Universität Darmstadt*

*bjarne.gregori@gmx.de*

**Elisabeth Dillies**
*Sorbonne Université*

*elisabeth.dillies@gmail.com*

**Quentin Delfosse**†
*Technische Universität Darmstadt*
*Google Intrinsic AI Research*

*quentin.delfosse@tu-darmstadt.de*

**Kristian Kersting**
*The Hessian Center for Artificial Intelligence (hessian.AI)*
*Centre for Cognitive Science*
*German Research Center for Artificial Intelligence (DFKI)*
*Technische Universität Darmstadt*

*kersting@tu-darmstadt.de*

**Reviewed on OpenReview:** *https://openreview.net/forum?id=7BFbso4B3R*

## Abstract

Pixel-based reinforcement learning agents often exploit spurious visual correlations, leading to brittle policies that fail under minor visual perturbations. We systematically investigate spatial grounded semantic channel representations, often called Feature Maps, Planes, or Object Channels, as a representation design principle for reducing shortcut learning. Object channels map detected entities into binary tensors aligned with the original coordinate frame, preserving compatibility with standard RL backbones without architectural modifications. Specifically, through systematic evaluation in Atari environments under controlled perturbations, we demonstrate that such channel representations substantially improve zero-shot robustness to distribution shifts while maintaining competitive in-distribution performance. We analyze the abstraction–fidelity trade-off and show that combining object channels with raw pixels improves robustness and sample efficiency compared to pure pixel-based approaches. The experimental results indicate that spatially grounded object-based encodings offer a practical mechanism for bridging pixel- and object-centric RL.

---

\*These authors contributed equally.
†Work done while at Technische Universität Darmstadt; now at Google Intrinsic AI Research.

# 1 Introduction

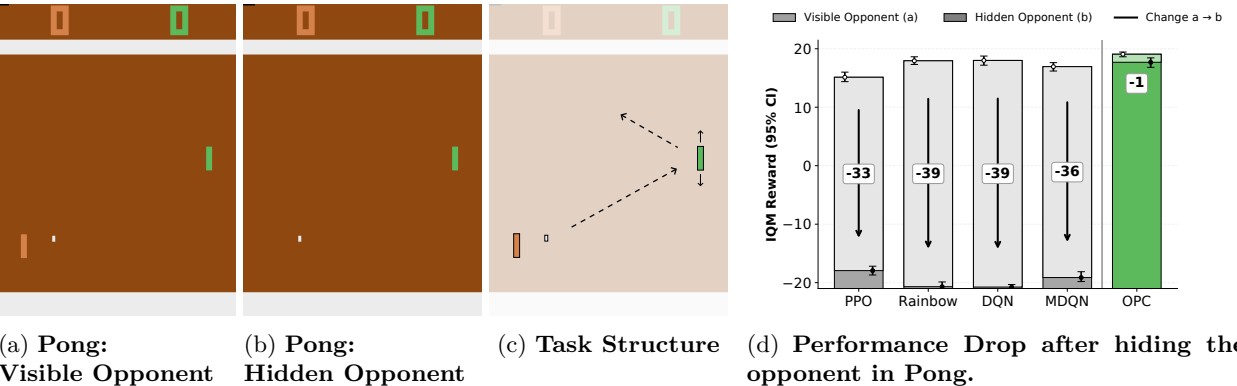

(a) **Pong: Visible Opponent**

(b) **Pong: Hidden Opponent**

(c) **Task Structure**

(d) **Performance Drop after hiding the opponent in Pong.**

Figure 1: **Object-Pixel Channels limit the performance drop in RL due to shortcuts.** Deep RL agents often rely on spurious visual cues rather than causal game mechanics. When confronted with perturbations that do not change the underlying task structure, such as **(b)**, their performance drops significantly. OPC prevent this by adding an object-based bias.

Pixel-based reinforcement learning (RL) agents frequently exploit *spurious visual correlations*: shortcut features that are not causally related to task success but provide low-entropy signals during training. Such *shortcut learning* leads to brittle policies that fail under minor visual perturbations that do not alter the environment's underlying transition dynamics (Farebrother et al., 2018; Ilyas et al., 2019; Chan et al., 2020; Geirhos et al., 2020; Langosco et al., 2022; Hermann et al., 2024; Delfosse et al., 2024a). Figure 1 illustrates the canonical example *Hidden Opponent Pong* (Delfosse et al., 2024c). During standard training, the opponent's movement is tightly correlated with the ball trajectory. Many agents implicitly exploit this correlation as a predictive cue for ball direction. When the opponent paddle is hidden, policies collapse, even though identical physics and reward structure are preserved.

A natural response to this brittleness is to bias learning toward task-relevant entities rather than raw visual features. This idea is strongly aligned with cognitive science accounts of perception. Human visual processing is often described as proceeding from rapid, pre-attentive feature detection (Treisman, 1985) toward structured, entity-centered representations (Treisman & Gelade, 1980). Such abstractions form the basis of reasoning and planning (Baars, 1988; 2002; Bengio, 2017; Goyal & Bengio, 2022). Object-centric reinforcement learning aims to incorporate a similar inductive bias, encouraging agents to organize perception around entities rather than raw pixel intensities. However, many object-centric RL (OCRL) approaches introduce substantial architectural changes, including slot-based attention mechanisms (Locatello et al., 2020) or fully symbolic pipelines (Li et al., 2024). While expressive, these methods add training instability, computational overhead, and design complexity. Moreover, some object-centric abstractions discard the spatial inductive biases of convolutional neural networks (CNNs), such as locality and translation invariance, that are critical for efficient visual learning (LeCun et al., 1998; Bronstein et al., 2017; Dosovitskiy et al., 2021). Further analyses emphasize that abstraction is beneficial only when it preserves task-relevant structure, and harmful when it removes information required for planning or control (Ho et al., 2019; 2022; Abel, 2020). Prior work has demonstrated that augmenting pixel inputs with object masks can substantially improve learning in individual environments. In particular, Davidson & Lake (2020) show that adding semantic object segmentation masks as additional input channels improves performance and generalization in *Frostbite*, and analyzes how models utilize such object information. These preliminary results indicate that simple object-channel augmentations can serve as a useful inductive bias.

In this work, we build on this foundation but shift the focus to a more systematic evaluation of the robustness, abstraction, and scaling properties of spatially grounded semantic channel representations. Rather than proposing a new policy architecture, we treat semantic channels as a representation design principle and evaluate their behavior across multiple Atari environments under controlled perturbation regimes. Concretely, we study spatially grounded semantic channel representations, which we refer to as Object&Pixel-Channels

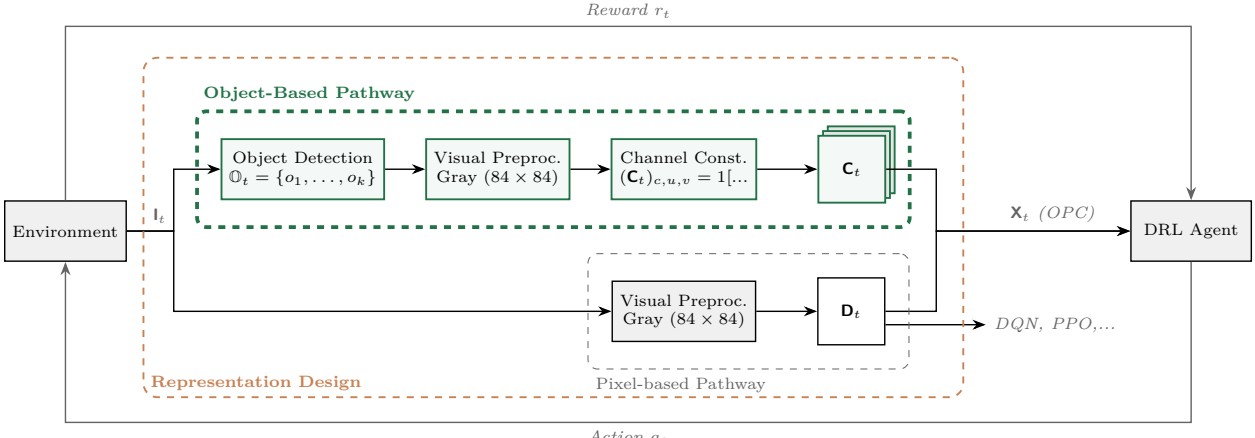

Figure 2: **The OPC Pipeline Architecture.** The environment provides raw frames $\mathbf{I}_t$, which are processed through two parallel pathways. The **Object-Based Pathway** constructs binary object channels $\mathbf{C}_t$. The **Pixel-based Pathway** generates standard downsampled grayscale observations $\mathbf{D}_t$. These streams are concatenated into a hybrid input $\mathbf{X}_t$, enabling the DRL agent to leverage both semantic abstraction and raw visual context.

(OPC), that map detected entities into a multi-channel binary tensor, with one channel per semantic object category (object channels) and one additional channel for the pixel observations (pixel channel). These object channels are aligned with the original coordinate frame, allowing standard convolutional backbones to process them using the same spatial inductive biases that underpin pixel-based RL. To isolate representation effects, we hold the RL algorithms and network architectures constant and vary only the input encoding. Using the `HackAtari` benchmark (Delfosse et al., 2024a), we evaluate zero-shot robustness under controlled perturbations, analyze structural failures arising from over-abstraction, measure scaling under fixed compute budgets, and quantify sensitivity to synthetic and learned detector noise.

**Contributions:**

- We systematically evaluate spatially grounded Object&Pixel-Channels (OPC) as an inductive bias for deep RL, isolating representation geometry from architectural effects.
- Across multiple Atari environments, we show that OPC improves zero-shot robustness to perturbations.
- We demonstrate how keeping pixels can prevent over-abstraction in object-based approaches.
- We characterize scaling behavior and sensitivity to detector noise, clarifying how representation geometry interacts with computation and imperfect perception.

The remainder of the paper is organized as follows. In Section 2, we describe the construction of OPC and their integration into standard convolutional RL agents. Section 3 presents controlled evaluations of performance, robustness, abstraction trade-offs, scaling behavior, and detection noise sensitivity. Section 5 situates our findings within the broader literature on object-centric reinforcement learning and representation learning. We conclude with a discussion of the implications and limitations of representation-driven inductive biases for visual control.

## 2 Object&Pixel-Channel Representations

A significant challenge in purely object-centric reinforcement learning is the dependency on high-fidelity semantic state information, which is often unavailable or prohibitively expensive to extract in complex visual domains. Fully symbolic abstractions, such as those requiring exact part-level segmentation or dense attribute vectors, can provide robustness to shortcut learning but frequently rely on "oracle" knowledge that does not generalize to unstructured environments (Delfosse et al., 2024c).

In general, an *object* is defined as a discrete, task-relevant entity. It is characterized not by its constituent pixels but by its persistent identity, spatial boundaries (bounding boxes), and its functional capacity to interact with other entities according to the underlying game logic. While objects are ideal units for reinforcement learning, true semantic objects are often latent and difficult to extract directly from raw pixels. In this work, we use sprites as a supplementary representation. Unlike abstract objects, sprites are tangible visual units that can be more easily identified by standard object detectors. A critical distinction must be made between these discrete objects and the *background*, which represents the environment's global spatial structure. While both may be task-relevant, backgrounds lack the localized, independent movement characteristic of sprites. Our objective is to investigate how lightweight, easily available object-centric information, specifically bounding boxes, can be used to enhance robustness while preserving spatial reasoning. Rather than proposing a new policy architecture, we focus on the design of representation.

While semantic channel representations are standard in structured domains like board games (Silver et al., 2016; Czech et al., 2020; 2024) and real-time strategy (RTS) games (Vinyals et al., 2017), where the state is explicitly defined, their role in the robustness of visual deep RL (DRL) remains underexplored. The concept of augmenting pixel inputs with object masks was evaluated by Davidson & Lake (2020) in the context of learning dynamics for the Atari game *Frostbite* and is also explored in Gmelin et al. (2023); Wang et al. (2023); Shi et al. (2024); Grooten et al. (2024); Lepert et al. (2025).

Our approach formalizes semantic channels as a principled representation design for mitigating the misalignment inherent in pixel-based DRL. Our architecture is shown in Figure 2. By systematically decoupling semantic entities from visuals across diverse environments, we demonstrate that spatially grounded object-based encodings act as a critical inductive bias that counters the model's tendency for shortcut learning. Consequently, our work provides a comprehensive characterization of the abstraction-fidelity trade-off, establishing semantic channels not merely as an add-on, but as a robust bridge that ensures *informational sufficiency* while maintaining the spatial priors of convolutional architectures. We formalize the construction of object channels by mapping object-entity data into a spatially grounded multi-channel tensor. This transformation preserves scene topology while enforcing semantic separation across channels. Let $\mathsf{I}_t \in \mathbb{R}^{3 \times H \times W}$ denote the raw RGB observation at time step $t$. We assume access to a set of detected objects $\mathbb{O}_t = \{o_1, \ldots, o_k\}$, while an object (or sprite) $o$ is defined as a spatially localized, task-relevant entity extracted from a frame $s \in \mathcal{S}$. Formally, in this work, an object is a tuple of structured properties

$$o = \langle l, (x, y), (w, h), \boldsymbol{f} \rangle, \tag{1}$$

with a categorical label $l_i \in \mathbb{C}$, a position $(x, y)$, and dimensions $(w, h)$ from which a bounding box can be deduced

$$\mathbb{B}_i \subset \{1, \ldots, H\} \times \{1, \ldots, W\}. \tag{2}$$

$\boldsymbol{f}$ describes additional object features that can be deduced depending on the extraction method, such as orientation, speed, or value. In this work, $\boldsymbol{f}$ will be ignored since the extraction of such features from sprites is expensive or impossible. From $\mathbb{O}_t$, we derive an object channel tensor $\mathbf{C}_t \in \{0, 1\}^{|\mathbb{C}| \times H \times W}$, where each channel corresponds to a discrete semantic category. For category $l$ and spatial location $(u, v)$:

$$(\mathbf{C}_t)_{l,u,v} = \mathbf{1}\big[(u, v) \in \mathbb{U}_l\big], \qquad \text{where } \mathbb{U}_l = \bigcup_{i:\, l_i = l} \mathbb{B}_i. \tag{3}$$

This formulation decomposes the scene into task-relevant semantic layers (object channels), e.g., player, enemies, projectiles, while remaining exactly aligned with the original coordinate frame. Unlike vectorized symbolic abstractions that collapse spatial relationships into fixed-length representations (Locatello et al., 2020; Delfosse et al., 2024b;c; Matthews et al., 2024), semantic channel representations preserve spatial locality and relative geometry. Standard convolutional kernels can therefore extract relational features directly, without requiring explicit relational modules or graph constructions (Yang et al., 2018). From an inductive bias perspective, semantic channels provide sparse, semantically grounded signals that suppress shortcut-prone visual variance while maintaining compatibility with convolutional spatial priors.

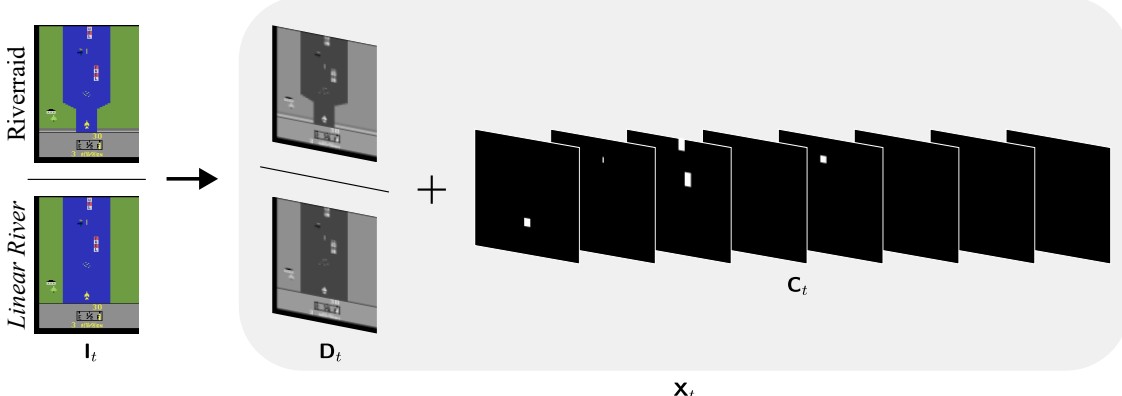

Figure 3: **Semantic Channel Decomposition in *Riverraid*.** This figure illustrates the OCP representation of task-relevant objects in the game Riverraid. While sprites, the underlying graphical bitmaps for the Player, Enemies, etc., are easily isolated into individual masks, the river is notably excluded. In this framework, the river is classified as a global spatial background structure rather than a discrete object. Illustrated are both the original *Riverraid*, as well as the *Linear River* modification of it.

## 2.1 Balancing Abstraction and Visual Fidelity

Purely symbolic abstractions, e.g., $\mathbf{C}_t$, can degrade performance in environments that require precise background geometry, such as the river boundaries in *Riverraid*. To study the trade-off between semantic abstraction and visual completeness, we combine these with a pixel-based representation $\mathbf{D}_t$. We refer to this combined representation as Object&Pixel-Channels (OPC), Following Mnih et al. (2015), we denote standard grayscale pixel observations as $\mathbf{D}_t \in \mathbb{R}^{1\times 84\times 84}$. $\mathbf{X}_t$ is constructed by concatenating this pixel channel with the object channels $\mathbf{C}_t$:

$$\mathbf{X}_t = [\mathbf{D}_t; \mathbf{C}_t] \in \mathbb{R}^{(|\mathbb{C}|+1)\times 84\times 84}, \tag{4}$$

where the concatenation occurs along the channel dimension. This design allows us to treat the pixel channel as a "visual baseline" and the object channels as "logical anchors". A visualization of this representation is shown in Figure 3.

OPC does not introduce a new optimization objective; the standard RL objectives are used unchanged, with $\mathbf{X}_t$ replacing $\mathbf{D}_t$ as the sole input modification. The connection between this representational choice and robustness can be stated formally.

**Proposition 1** (Perturbation Invariance of Object Channels). *Let $\mathbf{C}_t$ be the object channel tensor constructed from frame $\mathbf{I}_t$ via Equation (3). For any perturbation $\psi\colon \mathcal{I} \to \mathcal{I}$ that leaves all object bounding boxes $\mathbb{B}_i$ and category labels $l_i$ unchanged for all $i$, $\mathbf{C}_t(\psi(\mathbf{I}_t)) = \mathbf{C}_t(\mathbf{I}_t)$.*

*Proof.* Defined in Equation (3), $\mathbb{U}_l$ only depends on the bounding boxes $\mathbb{B}_i$ and labels $l_i$. Since Proposition 1 defined $\mathbb{B}_{i,\psi(\mathbf{I}_t)} = \mathbb{B}_{i,\mathbf{I}_t}$ and $l_{i,\psi(\mathbf{I}_t)} = l_{i,\mathbf{I}_t}$, again with Equation (3) follows that $(\mathbf{C}_t)_{l,u,v}$ must identical for all $(l, u, v)$ as well. □

By Proposition 1, any spurious visual feature that does not affect object layout is suppressed in $\mathbf{C}_t$, directly explaining the zero-shot robustness observed in Section 3. Beyond rendering-level shifts, we further evaluate robustness to perturbations that directly affect $\mathbf{I}_t$ itself, including hidden object (Pong) or localization jitter (SpaceInvaders).

In this context, our work evaluates the capacity of spatially grounded semantic features to override the inherent bias of deep RL agents to exploit low-level shortcuts. By providing a representation in which entities are pre-segmented and spatially localized, we effectively lower the "cost" of learning robust policies.

## 2.2 Architectural Compatibility and Object Detection

Since the OPC tensor $\mathbf{X}_t$ shares the same spatial topology as classical observations $\mathbf{D}_t$, it can be integrated into standard convolutional RL backbones—such as DQN (Mnih et al., 2015), PPO (Schulman et al., 2017), or Rainbow (Hessel et al., 2018)—with minimal modification to the first convolutional layer to accommodate additional input channels (cf. (Davidson & Lake, 2020)). No other architectural changes are required. Across all experiments, the learning algorithm, optimization procedure, and network structure remain fixed; only the input encoding varies. This enables a controlled isolation of representation effects. The mapping from raw pixels to object channels is detection-agnostic and modular. Object information may originate from simulator annotations, rule-based color segmentation (Davidson & Lake, 2020), RAM extraction (Delfosse et al., 2024b), or learned vision models such as YOLO (Redmon et al., 2016) or RT-DETR (Zhao et al., 2024). Regardless of the source, detected entities are projected onto the $H \times W$ spatial grid to construct $\mathbf{X}_t$ before being consumed by the RL backbone. This modular design creates a clean separation between perception and control: the detection module determines which entities are available, while the representation defines how that information is spatially organized for learning. As a result, representation geometry and detection quality can be analyzed as independent factors. In our experiments, we focus primarily on the representation design, while later sections examine robustness under imperfect detection to assess sensitivity to perception noise.

## 3 Experimental Evaluation

Our evaluation investigates how spatially grounded semantic channel representations influence performance, robustness, and spatial reasoning in reinforcement learning agents. By holding the optimization algorithm and network architecture fixed, we isolate the effect of representation and examine whether semantic anchors improve generalization under visual distribution shifts. Specifically, we address the following research questions:

**(Q1) Performance and Robustness:** How does encoding object information affect in-distribution performance and zero-shot generalization under visual perturbations?

**(Q2) Abstraction–Fidelity Trade-off:** Does retaining pixel-based information mitigate the geometric information loss in purely object-based representations while preserving robustness?

**(Q3) Detector Noise Sensitivity:** How does performance degrade under stochastic noise or imperfect object detection?

**(Q4) Computational Efficiency and Sample Complexity:** To what extent does the increased dimensionality of $\mathbf{X}_t$ Impact wall-clock training, throughput, and efficiency?

**Experimental Setup**

Our experimental framework is designed to decouple the input representation from optimization, enabling a focused evaluation of object-centric inductive biases. To distinguish between "understanding" and "shortcut learning," we utilize the `HackAtari` benchmark (Delfosse et al., 2024a). We implement a *Train-on-Clean, Test-on-Modified* protocol: agents are trained exclusively on standard, non-perturbed Atari games (i.e., the original ROMs) and subsequently evaluated zero-shot on variants with altered elements (e.g., background textures, color palettes, etc.). This allows us to quantify the *Generalization Gap*, the difference between in-distribution and zero-shot modified performance, thereby revealing the extent to which a policy is anchored to invariant object semantics rather than volatile pixel data (cf. Section 3.1). The used variants are explained in more detail in Appendix H. A central tenet of this work is the functional decoupling of the *perception module* (Object Detection, cf. Figure 2) from the policy. By initially utilizing an oracle for extraction (Delfosse et al., 2024b), we ensure that our analysis of representation design is not confounded by the performance of specific vision models. For more details about the extraction process, see Appendix G. We subsequently bridge this gap in Section 3.3 by re-introducing perceptual uncertainty via stochastic noise and integration with more common detectors, in our case YOLO12 (Redmon et al., 2016; Tian et al., 2025) and RT-DETR (Zhao et al., 2024).

Table 1: **Hybrid approaches outperform pixel and object-based approaches.** Performance Comparison on default Atari Environments. Interquartile Mean (IQM) episodic rewards with 95% stratified bootstrap confidence intervals (CIs) across 3 seeds. Bold indicates the highest performance per game.

| | Pixel-based | Hybrid | | Object-based |
|---|---|---|---|---|
| **Game** | PPO ($\mathbf{D}_t$) | OPC ($\mathbf{X}_t$) | SemVec | SCoBots (DT) |
| Amidar | 551 [521,570] | **903** [840,945] | 209 [184,232] | 178 [160,197] |
| Bowling | 65 [64,67] | 60 [60,60] | 62 [60,65] | **156** [120,196] |
| Boxing | 95 [94,96] | **96** [95,98] | 93 [90,96] | 82 [75,88] |
| Breakout | 131 [84,183] | **280** [209,338] | 37 [30,43] | — |
| Freeway | 32 [31,33] | **33** [33,33] | 31 [31,31] | 27 [26,28] |
| Frostbite | **300** [292,306] | 296 [288,306] | 265 [258,390] | — |
| MsPacman | 3001 [2803,3396] | **5931** [5424,6571] | 1919 [1406,2631] | — |
| Pong | 15 [14,16] | **19** [19,19] | **19** [18,20] | 16 [14,17] |
| Riverraid | 7668 [7279,8046] | **9753** [9523,10007] | 3306 [3069,3567] | 2441 [2321,2568] |
| SpaceInvaders | 744 [703,796] | **1594** [1285,1967] | 358 [308,419] | 389 [328,460] |

For performance, efficiency, and scalability, we compare against two baseline families: (i) *pixel-based* agents, including PPO (Schulman et al., 2017), DQN (Mnih et al., 2015), MDQN (Vieillard et al., 2020), and Rainbow (Hessel et al., 2018); and (ii) *symbolic* agents that consume flattened object semantic vectors (Delfosse et al., 2024b;c) or learn decision trees from them (SCoBots (Delfosse et al., 2024c)). All pixel-based agents share a standard Nature-CNN backbone (Mnih et al., 2015) and identical hyperparameters. The semantic vectors use a large MLP network of a similar size to the CNNs, while SCoBots are trained using the hyperparameters and network from the original work. The OPC networks require a modification only to the first convolutional layer to accommodate the $1 + |\mathbb{C}|$ input channels. Following the standard protocols for Atari (Machado et al., 2018) and Huang et al. (2022b) for training, we employ sticky actions, buffer the last 4 frames, and apply frame skipping. Training was conducted for 40 million frames across three seeds. Frames are resized and grayscaled to $84 \times 84$. Performance is quantified using the *Interquartile Mean (IQM)* with 95% stratified bootstrap confidence intervals and *Human-Normalized Scores (HNS)* as proposed by Agarwal et al. (2021), thereby ensuring a robust, game-agnostic comparative analysis. Human evaluation scores are taken from Badia et al. (2020). Details about setup, hyperparameters, and agents are in Appendix D. For Metrics, see Appendix E; extended results are in Appendix F.

### 3.1 Performance and Robustness to Perturbations

We first establish a performance baseline in unmodified Atari environments to ensure that introducing OPC does not degrade the underlying agent's capabilities. As shown in Table 1, using OPC consistently leads to high human-normalized scores across the suite, outperforming standard pixel-based PPO. These results indicate that spatially-grounded OPC provide a sufficient state representation for high-level control. Critically, the results suggest that semantic anchors can act as a catalyst for performance rather than a constraint, establishing a strong baseline for subsequent robustness evaluations.

**Zero-Shot Robustness Under Perturbations.** To assess the degree of causal invariance in the learned policies, we conduct zero-shot evaluations on `HackAtari` variants that introduce significant distribution shifts while maintaining identical transition dynamics. As illustrated in Figure 4, extended in Appendix F, standard pixel-based baselines exhibit a catastrophic generalization gap under these shifts, confirming a reliance on spurious visual correlations. For instance, in *Boxing*, pixel-based agents suffer total policy collapse due to altered colors, whereas the OPC agent maintains near-perfect performance. More strikingly, in the *Hidden Opponent Pong* variant, where the opponent paddle is visually removed (cf. Figure 1), pixel-based and symbolic vector agents fail, indicating reliance on rendered opponent cues as a shortcut for trajectory prediction. However, the OPC agent maintains its performance, suggesting that the policy is grounded in relational ball–paddle dynamics rather than opponent visibility.

Our results confirm that spatially grounded semantic channel representations significantly enhance an agent's robustness to visual style perturbations while preserving competitive asymptotic performance. By projecting semantic entities, i.e., objects, into a topological grid, we provide a geometric inductive bias that allows the policy to remain invariant to non-causal shifts in the input distribution. Critically, these findings demonstrate that the choice of representation geometry alone can meaningfully reduce the chance of shortcut learning, providing a robust scaffold for decision-making without requiring specialized architectural modifications to the reinforcement learning backbone.

## 3.2 The Abstraction–Fidelity Trade-Off

While semantic abstraction ($\mathsf{C}_t$) minimizes space by filtering out non-entity variance, it introduces an informational bottleneck if the underlying object ontology is incomplete. We characterize this as a transition to a functionally incomplete or partially observable MDP, where the mapping $f : \mathsf{I}_t \rightarrow \mathsf{C}_t$ discards state information necessary to satisfy the transition dynamics $P(s_{t+1}|s_t, a_t)$. In practice, this bottleneck can stem from many sources: limitations of the underlying detector (e.g., failing to extract an entity) or an ontological design choice (e.g., omitting an object type). While these are empirically difficult to disentangle in our current setup, the downstream

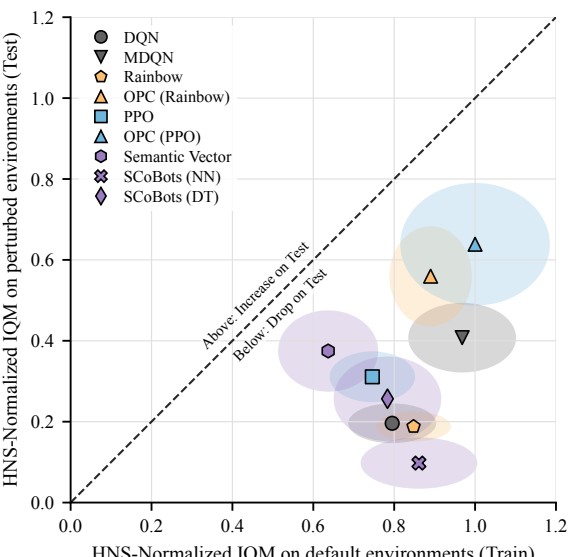

Figure 4: **Zero-Shot Generalization under Perturbations.** Each point shows the HNS normalized IQM return of an agent on clean training environments (x-axis) versus perturbed test environments (y-axis). Points closer to the diagonal indicate greater robustness to environmental changes. The desired outcome is the upper-right corner, corresponding to high performance in both settings. The ellipses indicate the 95% confidence interval. Game-specific performance results are in Appendix F.

effect remains the same. In *Riverraid* riverbanks define the navigable space, yet they are not regarded as objects by OCAtari and as such are not detected. Consequently, purely symbolic agents suffer from distinct configurations (e.g., safe passage vs. imminent collision with terrain) that appear identical in the state space—leading to a significant performance collapse (cf. Table 2). This failure mode highlights a fundamental trade-off: while object-centric abstractions filter out visual noise, they risk discarding information required for valid policy mapping. We refer to this consequence as "over-abstraction", the loss of task-relevant data caused by a representation that is drawn too narrowly, independent of its reason.

The OPC architecture addresses the risk of ontological incompleteness by preserving a high-fidelity pixel channel alongside semantic abstractions. As demonstrated in our *Riverraid* evaluations, the OPC agents maintain consistent performance across both vanilla and modified variants. This resilience indicates that pixel-level geometry functions as a spatial fallback, compensating for gaps in the semantic object schema, while also increasing the robustness to, e.g., color changes, compared to pixel-based approaches (cf. Figure 4). However, reintroducing pixel grounding also partially reintroduces sensitivity to visual shortcuts, making hybridization a trade-off rather than a strictly dominant solution.

These results raise a fundamental question for representation design: What qualifies as an object? While structures like the river in Riverraid are global spatial backgrounds rather than discrete entities, they remain strictly task-relevant. Our findings highlight that defining the scope of an abstraction, deciding what to include and what to omit, is a non-trivial design choice. This reveals a central tension in object-centric reinforcement learning: increasing abstraction improves invariance but risks eliminating the perceptual cues necessary for high-fidelity planning. The OPC formulation provides an efficient middle ground, preserving convolutional spatial priors while injecting structured semantic bias. We thus demonstrate that representation geometry must optimize for *perceptual sufficiency* by balancing logical invariance with geometric fidelity. More details are provided in Section F.1.

Table 2: **Over-Abstraction in *Riverraid*.** IQM performance on *Riverraid* and two variants. Values in parentheses denote percentage change relative to the original score. Bold indicates stable performance ($|\Delta| \leq 10\%$) or improvement ($\Delta > 0$). Confidence intervals are reported in Appendix F.

| Game | PPO | | | Baselines | | | |
|---|---|---|---|---|---|---|---|
| | DQN-like ($\mathbf{D}_t$) | Object Channels ($\mathbf{C}_t$) | OPC ($\mathbf{X}_t$) | DQN | MDQN | Rainbow | SemVec |
| Riverraid | 7668 | 8020 | 9786 | 9202 | 8633 | 3009 | 3306 |
| Color Change | 441 (-94%) | **8263 (+3%)** | **9644 (-1%)** | 285 (-97%) | 868 (-90%) | 509 (-83%) | **3525 (+7%)** |
| Linear River | **6932 (-10%)** | 1969 (-75%) | **9634 (-2%)** | **10901 (+18%)** | **11568 (+34%)** | **5018 (+67%)** | 356 (-89%) |

| Detector | Freeway | Amidar | Pong |
|---|---|---|---|
| OCAtari REM | $33_{[33,33]}$ | $903_{[840,945]}$ | $19_{[19,19]}$ |
| OCAtari VEM | $33_{[33,33]}$ | — | $20_{[19,20]}$ |
| YOLO11n | $33_{[32,33]}$ | $702_{[598,771]}$ | $18_{[18,19]}$ |
| YOLO11l | $32_{[32,33]}$ | $639_{[541,746]}$ | $18_{[16,18]}$ |
| YOLO12n | $33_{[32,33]}$ | $653_{[586,727]}$ | $18_{[18,19]}$ |
| YOLO12l | $32_{[32,33]}$ | $578_{[492,647]}$ | $16_{[13,18]}$ |
| RT-DETR | $15_{[14,15]}$ | $46_{[33,59]}\downarrow$ | $16_{[14,17]}$ |

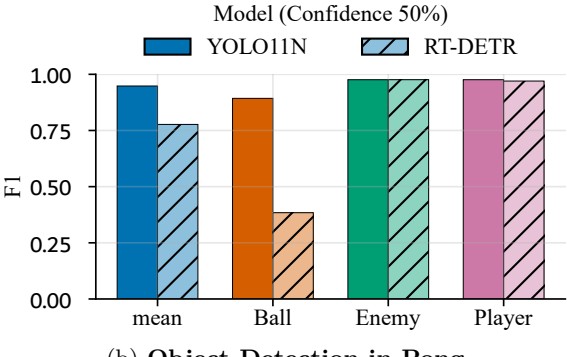

(a) **Performance of Object-Detectors on Atari**          (b) **Object Detection in Pong**

Figure 5: **Modern object detectors can reliably substitute perfect object annotations when detection quality is adequate.** Across several games, YOLO-based detectors closely match the OCAtari REM baseline, indicating that oracle-based object extraction can be replaced. Performance collapses only when critical objects are consistently missed, emphasizing that robustness depends on preserving task-relevant structure. Grey values indicate a collapse larger than an order of magnitude. More experiments and mAP scores in Section F.4.

## 3.3 Robustness to Perception Noise

While OPC provides significant robustness, it introduces a structural dependency on the fidelity of the upstream perception module. To evaluate the importance of object quality, we subjected the `OCAtari` oracle to controlled synthetic noise—including localization jitter, object dropout, and semantic misclassification at probabilities $p \in \{2\%, 5\%, 10\%\}$. Results reveal a stark contrast in noise tolerance based on task complexity. In *Freeway*, which relies on sparse entities and highly redundant visual cues, performance remains near-optimal even when 10% of objects are missing. However, *Breakout* and *Amidar* exhibit high sensitivity to "blind spots" in the object stream. Specifically, *Breakout* policies trained on low noise (2%) suffer catastrophic failure when tested on 10% noise, dropping by 70% when omitting the pixel channels (cf. Section F.4).

Evaluations using learned detectors (YOLO11/12 and RT-DETR) show that aggregate detection metrics, such as Mean Average Precision (mAP), are good indicators of downstream policy success. Figure 5 shows that agents supported with YOLO achieve high scores when the detection quality is high. However, when object detection becomes more difficult, e.g., in Pong, or when the underlying detection RT-DETRs perform poorly, the agent's performance decreases. Combining object channels with a pixel representation mitigates this problem, enabling the agent to handle such detection errors (cf. Section F.4).

Our results demonstrate that representation geometry and perception reliability must be co-designed for robust deployment. While modular separation enables clean ablations, it exposes the limitations of a strictly feed-forward perception-to-policy pipeline. Future work should explore mechanisms that promote the temporal consistency of task-relevant entities and integrate perceptual uncertainty directly into policy learning.

| Game | #Chn. | PPO | OPC |
|---|---|---|---|
| Freeway | 3 | **33** [33,33] | **33** [32,33] |
| Boxing | 5 | **93** [91,94] | **93** [91,95] |
| Pong | 5 | 16 [15,17] | **19** [18,20] |
| Breakout | 6 | 131 [86,200] | **221** [149,304] |
| Amidar | 7 | **555** [492,609] | 458 [442,477] |
| MsPacman | 7 | 2515 [2187,2899] | **5066** [4670,5529] |
| SpaceInvaders | 8 | 707 [631,804] | **1048** [879,1275] |
| Riverraid | 10 | **7609** [7169,8000] | 4979 [3999,5753] |
| Frostbite | 12 | 281 [280,291] | **289** [285,295] |

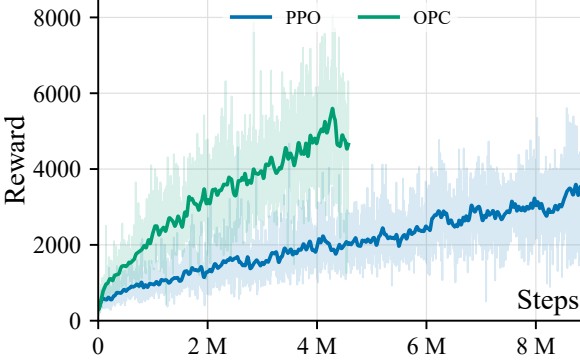

(a) **Evaluation under fixed wall-clock time constraints.**    (b) **90-minute learning curve on MsPacman.**

Figure 6: **Efficiency and scaling properties of semantic channel representations.** (a) Under a fixed 90-minute wall-clock budget, OPC agents frequently match or exceed pixel-only PPO despite processing fewer total transitions, as the semantic inductive bias compensates for reduced throughput. (b) Exemplary 90-minute learning curve on MsPacman. Full learning curves across all nine environments and GPU memory footprint data are provided in Appendix F.

## 3.4 Scalability and Sample Efficiency

Object channels increase input dimensionality proportionally to the number of object categories $|\mathbb{C}|$, yielding tensors of size $((|\mathbb{C}|+1) \times 84 \times 84)$. This primarily affects the first convolutional layer, leading to a decrease in the number of environment steps per second compared to pixel-only agents. Empirical measurements (cf. Figure 6) confirm approximately linear scaling in wall-clock cost as channel depth increases. While this additional cost is measurable, it must be evaluated in light of learning efficiency. Prior work by Davidson & Lake (2020) observed that augmenting pixel inputs with object masks accelerated learning in *Frostbite*, suggesting that semantic channels reduce the burden on convolutional filters to discover entities from raw pixels, thereby accelerating early-stage representation learning. Our multi-environment evaluation extends this observation: under fixed wall-clock budgets, OPC agents frequently achieve comparable or superior IQM performance relative to pixel-only PPO (cf. Figure 6a), despite processing fewer total transitions within the budget. This indicates that the semantic inductive bias can improve sample efficiency sufficiently to offset reduced throughput in many environments.

However, in environments with larger object vocabularies or high channel counts (e.g., Riverraid), the increased dimensionality can reduce the net efficiency gains from abstraction. Moreover, purely symbolic approaches can achieve higher raw throughput by bypassing spatial convolutions altogether, though often at the cost of geometric inductive bias and robustness. Thus, semantic channel representations occupy a middle ground: more computationally intensive than flattened semantic vectors, but structurally aligned with convolutional spatial priors. The practical value of OPC lies not in eliminating computational cost, but in shifting the optimization burden from discovering objects to reasoning over them. When object sets remain moderate in size, this shift yields favorable trade-offs between efficiency and robustness.

## 4 Discussion

**Spatial Grounding as a Robust Inductive Bias.** Our results suggest that the primary advantage of semantic channel representations over traditional symbolic ones, such as semantic vectors, lies in the preservation of topological structure. While purely symbolic approaches often discard the spatial context required for localized reasoning, OPC maintains a direct alignment with the convolutional layers' receptive fields. This spatial grounding effectively mitigates "shortcut learning". By explicitly decoupling entity detection from policy logic via object channels, we achieve a representation that is not only robust to visual distribution shifts but also naturally suited for tasks that require precise spatial coordination.

**Bridging the Perception Gap.** A fundamental challenge for deploying semantic channel representations in non-synthetic settings is the dependency on reliable object detection. While our experiments use oracle extraction to isolate representation effects, we observe a performance gap when transitioning to learned, imperfect detectors such as RT-DETR or YOLO, which is linked to their detection performance. The modular separation between perception and control means that detection models can be replaced without retraining the policy. Whether stronger foundation models such as Grounding DINO (Liu et al., 2024) or SAM3 (Carion et al., 2026) can further improve downstream performance remains an open question; both require domain-specific fine-tuning on Atari frames and do not generalize zero-shot to this visual domain, similar to YOLO (cf. Appendix D.4), which we leave as a direction for future work. To further bridge this gap, future work should explore co-training perception and policy to provide task-relevant feedback signals, or incorporate uncertainty-aware representations that treat detection confidence scores as additional input channels.

**Mitigating Over-Abstraction via Hybrid Architectures.** A critical observation in our evaluation is the risk of "over-abstraction" inherent in purely object-centric systems. In environments like *Riverraid*, omitting non-entity background features (e.g., river boundaries) led to catastrophic failure for pure object-based agents. The success of OPC indicates that pixels and object channels provide complementary signals: the object channels offer a high-level semantic bias that guides entity interaction, while the pixel channels act as a "visual safety net" for grounding the agent in static environment geometry. This hybrid architecture achieves a balance, retaining the high-fidelity context of CNNs without sacrificing the visual robustness of symbolic abstractions.

**From Visual to Structural Generalization.** While OPC agents demonstrate near-perfect robustness to visual perturbations (e.g., color and sprite changes), they remain sensitive to fundamental structural shifts, such as changes in maze layout geometry. The performance decline observed in *MsPacman* (see Appendix F) highlights a clear distinction between *visual robustness* and *structural generalization*. Although OPC ensures that the agent "recognizes" entities regardless of their rendering, they do not inherently enable the agent to "reason" about unfamiliar topologies. This suggests that representation can control what information is available to the policy, but not how the policy reasons over novel configurations. Achieving true level-agnostic navigation likely requires integrating complementary mechanisms such as graph neural networks for relational reasoning or meta-learning across environment layouts.

**Design Principles for Object-Centric Representations.** Our analysis of the generalization gap suggests three core design principles for building robust RL agents: (1) Prioritize Spatial Grounding: Semantic information should be encoded in the input representation, e.g., as spatially grounded channels, rather than flattened symbolic vectors. Preserving the coordinate-frame alignment allows the agent to leverage convolutional priors for relational reasoning, which we found critical for zero-shot generalization. (2) The Necessity of Hybridity: Pure abstraction often comes at the cost of "perceptual blindness" to task-relevant background geometry. Combining both pixels and semantic information should be the default, as the pixel channel provides the necessary geometric context that discrete object sets may overlook. (3) Build representation that matters: The goal of abstraction should not be to minimize the input size, but to align the representation with the underlying task's structure. "Seeing right" in robust RL involves capturing the invariant spatial relationships between entities. As demonstrated by our studies, focusing solely on object identities is insufficient; agents must maintain a bridge between high-level semantics and low-level visual evidence to achieve more flexibility. A promising direction for future work is the systematic evaluation of ontology granularity: merging categories with similar behavioral roles into shared channels offers a scalable path forward.

## 5 Related Work

**Shortcut Learning and Structural Robustness.** Deep RL agents trained from raw pixels often rely on superficial visual correlations rather than task-relevant causal structure. This phenomenon, termed *shortcut learning*, leads to brittle policies that collapse under minor distribution shifts (Farebrother et al., 2018; Ilyas et al., 2019; Chan et al., 2020; Langosco et al., 2022; Geirhos et al., 2020; Hermann et al., 2024; Delfosse et al., 2024a).

Existing mitigation strategies fall into two categories: (i) *extrinsic* methods, such as domain randomization (Tobin et al., 2017) and image augmentations (Laskin et al., 2020; Yarats et al., 2021), which increase data diversity to force invariance; and (ii) *algorithmic* methods utilizing auxiliary objectives for invariant feature embeddings (Zhang et al., 2021; Bertoin et al., 2022). While effective, these approaches operate on unstructured pixel grids where semantic entities are entangled with non-causal style features. In contrast, we investigate robustness as a property of representation, projecting detected entities into spatially-grounded object channels to decouple object mechanics from visual rendering.

**Inductive Bias and State Abstraction.** Robustness is fundamentally tied to inductive bias and state abstraction. Cognitive science suggests that humans prioritize structured entities over raw sensory detail (Treisman & Gelade, 1980; Kahneman, 2011; Baars, 1988; Bengio, 2017; Ho et al., 2022; Goyal & Bengio, 2022). In RL, *state abstraction* formalizes this through a mapping $\phi : \mathcal{S} \to \mathcal{Z}$ that aggregates states based on task-relevant invariants while preserving optimal policy value (Li et al., 2006; Abel et al., 2016; Abel, 2020). However, $\phi$ introduces a critical trade-off: discarding non-causal variation improves generalization, but over-abstraction can omit information essential for control (Ho et al., 2019; 2022). Unlike purely symbolic abstractions, spatially grounded object channels preserve spatial topology, maintaining geometric fidelity for spatial reasoning while providing semantic invariance against visual distribution shifts.

**Invariance, Masking, and Representation.** The benefits of maintaining such compatibility are further evidenced by research in visual invariance and masking. Improving generalization in RL has traditionally relied on injecting pixel-level invariances via data augmentation (e.g., RAD, DrQ) (Laskin et al., 2020; Yarats et al., 2021) or guiding policies toward salient regions using soft attention and reward-driven masking (Mott et al., 2019; Bertoin et al., 2022; Grooten et al., 2024). More recently, the use of segmentation masks—often extracted via foundation models like SAM—has enabled policies to transfer across visually diverse domains by filtering out non-essential textures (Wang et al., 2023; Shi et al., 2024; Lepert et al., 2025). However, the majority of these works focus on *mask extraction*, treating the perception bottleneck as the primary challenge. Our work diverges from this trend by investigating the *geometry* of the resulting representation. While symbolic approaches often compress objects into flattened semantic vectors, they frequently discard the topological structure necessary for spatial reasoning. By contrast, we examine how spatially grounded, multi-channel object masks serve as a critical inductive bias, bridging the gap between high-level symbolic abstraction and the spatial grounding required for robust policy learning.

**Object-Mask Augmentation and Channel Encodings.** Spatially grounded object channel augmentations have been explored in structured domains (board games, RTS) (Silver et al., 2016; Vinyals et al., 2017; Czech et al., 2020; 2024). While in board games, channels are often binary layers for specific pieces; in Atari, these channels must handle varying entity counts and overlapping coordinates. Davidson & Lake (2020) augment pixel observations with segmentation masks in Frostbite, introducing variants that concatenate object masks as input channels, showing accelerated learning and improved robustness within a single environment. Our work extends this foundation by treating semantic channels as a representation design principle for systematic multi-environment evaluation. We provide: (i) controlled evaluation across environments using HackAtari perturbations, (ii) analysis of the abstraction-fidelity trade-off, (iii) scaling behavior under fixed compute budgets, and (iv) robustness analysis under synthetic and learned detector noise.

This characterizes when and why semantic channel representations succeed or fail, providing practical guidance beyond single-environment demonstrations.

## 6 Conclusion

In this work, we conducted a systematic study of spatially grounded semantic channel representations as an inductive bias for visual reinforcement learning. By mapping detected entities into object channels aligned with convolutional spatial priors, we isolate representation geometry as the primary experimental variable. Across multiple Atari environments, semantic channel representations substantially reduce shortcut learning and improve zero-shot robustness to visual perturbations while remaining fully compatible with standard deep RL setups.

Our evaluation reveals a central trade-off between semantic abstraction and geometric completeness. Object channels without pixel information increase invariance by filtering spurious visual correlations, but may suffer from ontological incompleteness when task-relevant background structure is not modeled explicitly. OPC mitigates this risk by grounding object channels in raw pixels, acting as a safety net under structural variation and perception noise. Robustness, therefore, depends not on maximizing abstraction but on balancing invariance with spatial fidelity.

Although semantic channel representations increase input dimensionality linearly with the number of object categories, their semantic inductive bias often compensates for reduced throughput via improved learning efficiency. Under fixed wall-clock budgets, OPC agents frequently match or exceed pixel-only performance despite processing fewer total transitions. This positions semantic channel representations as a practical middle ground between raw pixels and flattened symbolic vectors.

Importantly, visual robustness does not imply structural generalization. Performance degradation under unseen layout changes highlights a clear boundary between invariance to rendering and reasoning over novel environment topologies. Furthermore, semantic channel representations introduce a dependency on object ontologies and upstream perception quality. Future work should investigate learned abstractions, adaptive object schemas, and tighter integration of perception and policy to move beyond visual robustness toward broader structural transfer.

Overall, our findings demonstrate that representation geometry alone can strongly influence robustness, efficiency, and shortcut-prevention in deep reinforcement learning. Spatially aligned semantic channel representations provide a principled and computationally tractable mechanism for injecting object-centric bias into standard RL pipelines, offering a concrete step toward more reliable systems.

## Author Contributions

JB and CD contributed equally to this work. They jointly conceived and designed the research, developed the experimental framework, conducted all evaluations, and led the writing of the manuscript. BG contributed to implementing the codebase and supported the experimental pipeline. QD and ED contributed to the writing and revision of the manuscript. KK provided supervision, conceptual guidance, and feedback throughout the research and on the manuscript.

## Acknowledgments

This research work has been funded by the Deutsche Forschungsgemeinschaft (DFG, German Research Foundation) under Germany's Excellence Strategy (EXC-3057/1 "Reasonable Artificial Intelligence", Project No. 533677015). Further, it was supported by the German Federal Ministry of Education and Research, the Hessian Ministry of Higher Education, Research, Science and the Arts (HMWK) within their joint support of the National Research Center for Applied Cybersecurity ATHENE (Project No. 50900401). The authors thank the reviewers and action editor of TMLR for their constructive feedback, which helped improve the clarity and scope of this work.

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

# A  Appendix

We provide the following elements in the appendix:

**Section B Usage of LLMs:** Disclosure of LLM assistance for linguistic refinement and LaTeX formatting.

**Section C Code:** Instructions for accessing the source code, environment wrappers, and scripts necessary to reproduce all experiments.

**Section D Extended Experimental Setup:** Comprehensive lists of hyperparameters, network architectures, and training configurations for all RL agents and object detectors.

**Section E Metrics:** Definitions and calculation procedures for the Normalized Score, Robustness Gap, Detection Quality, and Sample Efficiency metrics.

**Section F Extended Results:** Complete performance tables, learning curves, and statistical breakdowns for the full suite of Atari environments and perturbations, as well as our ablation into Craftax.

**Section G Object Detection with OCAtari:** Give a short introduction on how to extract objects from Atari via OCAtari.

**Section H HackAtari Variants:** Detailed descriptions and visual examples of the specific visual and structural modifications applied to each environment.

# B  Usage of LLMs

We acknowledge the use of tools such as ChatGPT, Gemini, DeepL Write, and Grammarly to assist with writing, grammar correction, and language polishing during the preparation of this manuscript.

# C  Code

All source code, experiment configurations, and processed evaluation data used in this work are available at `https://github.com/VanillaWhey/object-pixel-channels`. The repository includes implementations of OPC and training scripts for Atari environments to facilitate reproducibility.

# D  Extended Experimental Setup

To evaluate the effectiveness of object-centric abstraction in RL, we conduct a series of controlled experiments using the Atari Learning Environment (ALE). We benchmark OPC agents against conventional pixel-based RL methods, including Deep Q-Networks (DQN) (Mnih et al., 2015), Proximal Policy Optimization (PPO) (Schulman et al., 2017), Rainbow (Hessel et al., 2018), and MDQN (Vieillard et al., 2020), as well as OCAtari (Semantic Vector) (Delfosse et al., 2024b) and SCoBots (Delfosse et al., 2024c) as two object-centric representations with a focus on symbolic representation. The DQN and MDQN baseline models were taken from Gogianu et al. (2022), while the PPO, Rainbow, and Semantic Vector and SCoBots agents were trained by us. To test robustness, we evaluate the trained agents not only in their original environment but also under perturbations. These perturbations, derived from HackAtari (Delfosse et al., 2024a), include visual alterations (e.g., color changes, object recoloring), structural modifications (e.g., object displacement, swapped game elements), and gameplay variations (e.g., altered agent dynamics or opponent/enemy behavior). The metrics used are the human-normalized score (HNS) over aggregated game scores, using the interquartile mean (IQM). Calculations are done with rliable (Agarwal et al., 2021). The metrics are described in Section E.

### D.1 Environment Selection

We evaluate our framework across 10 Atari games, selected to balance reactive and strategic decision-making (Boxing vs. Freeway), environments where background information or object features are crucial (MsPacman), and some classics, such as Pong and Breakout. Overall, we wanted to include games of the following 5 categories with differences in difficulty:

- **Navigation and Avoidance**: Amidar, Freeway

- **Shooting and/or moving targets** SpaceInvader, Riverraid, Bowling

- **Paddle + Ball**: Pong, Breakout

- **Multi-Object Strategy** MsPacman, Frostbite

- **Reactive, Fast**: Boxing

This selection allows us to assess the adaptability of different representations across a spectrum of task complexities. Further, we used Craftax (Matthews et al., 2024) as yet another environment to test a subset of approaches.

### D.2 Training with Semantic Channels

As a basis for our experiments, we used PPO, which is a policy gradient algorithm widely used in deep RL that optimizes a clipped surrogate objective to balance exploration and stability (Schulman et al., 2017). Unlike value-based approaches, such as DQN, PPO directly learns an optimal policy distribution and is known for its sample efficiency and stable convergence properties. It is one of the most common architectures used in RL and, as such, is a good baseline and a good starting point for our experimental section. We trained PPO agents for the experiments using varying input representations (OPC, SemVec, SCoBots). This allowed us to isolate the trade-offs between abstraction strength and spatial reasoning capacity and evaluate our visual reasoning with object-centric attention. All agents are trained on the unmodified versions of these environments for 40 million frames, with hyperparameters adapted from Huang et al. (2022b) and listed in Table 3. We adhere to standardized implementation guidelines (Huang et al., 2022a) to ensure comparability and use the aforementioned CleanRL framework (Huang et al., 2022b) for training, as well as Stable-Baselines3 Raffin et al. (2021), for SCoBots. The object-centric representations, derived using OCAtari, selectively preserve task-relevant features while eliminating extraneous background information. Importantly, no fine-tuning is performed in perturbed environments, ensuring that generalization performance reflects the robustness of the learned representations rather than additional adaptation. Performance is evaluated using average episodic rewards across 10 games per seed, with three fixed seeds per experiment. All experiments are conducted on NVIDIA V100 GPUs, with training times averaging 1.5–5h across environments (cf. Figure 8).

### D.3 Training Rainbow

Rainbow is a value-based deep RL algorithm that integrates several improvements over the original DQN architecture, including prioritized replay, n-step returns, distributional value learning, and dueling networks (Hessel et al., 2018). As a strong, widely adopted baseline in Atari benchmarks, Rainbow serves as a useful addition to PPO for evaluating the effectiveness of semantic channel representations. We trained the Rainbow agents using the same set of input representations as in the PPO experiments. This setup allows us to assess whether the benefits of semantic channel representations generalize across different learning paradigms, namely, value-based and policy-based methods. All Rainbow agents were trained for 40 million environment frames using the implementation from the CleanRL framework (Huang et al., 2022b), with hyperparameters adapted to match standard evaluation settings, introduced by Dopamine (Castro et al., 2018) (see Table 4). As with PPO, the object-centric inputs are derived using OCAtari, and no fine-tuning is performed on perturbed environments. We, however, had to reduce the replay buffer size for OPC from $10^6$ to $10^5$ to improve training stability and address resource limitations. The reduction may reduce sample diversity and affect the stability of off-policy learning for Rainbow. Results for Rainbow should therefore be

interpreted as a lower bound on potential performance, and direct comparisons with the full-buffer Rainbow baseline should be made with this caveat in mind. Performance is evaluated using average episodic rewards over 10 episodes per seed, with three seeds per condition. All training was conducted on NVIDIA V100 GPUs. Training time per agent per seed ranged from 20 to 30 hours for most representations. This huge difference is mainly due to the implementation, which does not use parallel environments, unlike PPO. The total GPU cost of Rainbow training across all seeds and conditions was approximately 3,000 GPU hours. The complete hardware description is given in Table 5. Due to the high training time required for these experiments, we keep this a small ablation focused on network architecture and do not further investigate it. Results of these experiments can be found in Table 7.

### D.4 Fine-tuning Learned Object Detectors.

To evaluate the transition from oracle extraction to learned perception, we fine-tuned state-of-the-art YOLO11/12 and RT-DETR models from the Ultralytics suite[1] on Atari object distributions. Using OCAtari's REM annotations as ground truth, we constructed a dataset of 10 episodes per game ($\sim$20,000 frames) for three representative environments (*Pong, Freeway, Amidar*), utilizing COCO-pretrained weights and fine-tuning for 100 epochs with default data augmentation, following the best practices of Ultralytics[2]. The resulting models were integrated into the OPC pipeline to replace the oracle detector, allowing us to assess policy performance under realistic perception noise. Furthermore, we compared these deep learning approaches against the OCAtari's VEM, a lightweight, rule-based proxy that infers object bounding boxes directly from RGB frames via template matching, providing a non-oracle, vision-based baseline.

### D.5 Hyperparameters

We detail the hyperparameters used in our training in Tables 3, 4 and 6 to ensure reproducibility and consistency in our experiments. These settings were chosen based on prior literature.

Table 3: Key hyperparameters used for **PPO** training, following (Huang et al., 2022a) and (Huang et al., 2022b).

| Hyperparameter | Value |
|---|---|
| Seed | $\{0, 1, 2\}$ |
| Learning Rate ($\alpha$) | $2.5 \times 10^{-4}$ |
| Total Timesteps | $10^7$ |
| Number of Environments | 10 |
| Batch Size ($B$) | 1,280 |
| Minibatch Size ($b$) | 320 |
| Update Epochs | 4 |
| GAE Lambda ($\lambda$) | 0.95 |
| Discount Factor ($\gamma$) | 0.99 |
| Value Function Coefficient ($c_v$) | 0.5 |
| Entropy Coefficient ($c_e$) | 0.01 |
| Clipping Coefficient ($\epsilon$) | 0.1 |
| Clip Value Loss | True |
| Max Gradient Norm ($\|g\|_{\max}$) | 0.5 |

---

[1] `https://www.ultralytics.com/yolo`, accessed 2025-11-14.
[2] `https://docs.ultralytics.com/modes/train/`, accessed 2025-11-14.

Table 4: Key hyperparameters used for **Rainbow** training, following (Castro et al., 2018).

| Hyperparameter | Value |
|---|---|
| Total Environment Steps | $10^7$ |
| Replay Buffer Size | $10^6/10^5$ |
| N-step Returns | 3 |
| Value Distribution Atoms | 51 |
| Value Support Range | $[-10, 10]$ |
| Batch Size | 32 |
| Target Network Update Frequency | 8,000 |
| Optimizer | Adam |
| Learning Rate ($\alpha$) | $6.25 \times 10^{-5}$ |
| $\epsilon$-greedy Start | 1.0 |
| $\epsilon$-greedy End | 0.01 |
| $\epsilon$ Anneal Steps | 1,000,000 |
| Discount Factor ($\gamma$) | 0.99 |
| Min Replay Size | 1,600 |
| Update Frequency | Every 4 steps |
| Max Gradient Norm | 10.0 |
| Noisy Nets | False |
| Dueling Architecture | True |
| Prioritized Replay | True |
| Priority Exponent ($\alpha_{\text{prio}}$) | 0.5 |
| Priority Importance Sampling ($\beta_{\text{prio}}$) | $0.4 \to 1.0$ |

Table 5: **Hardware configuration** used for all experiments. We use an NVIDIA DGX system equipped with V100 GPUs and follow standardized training pipelines.

| Component | Specification |
|---|---|
| System | NVIDIA DGX v.5.1.0 |
| GPUs | $16 \times$ NVIDIA V100 32GB |
| CPUs | Intel Xeon Platinum 8174 |
| System Memory | 1.58 TB |
| Operating System | Ubuntu 20.04 LTS |
| CUDA Version | 12.4 |
| Python Version | 3.10 |

Table 6: Key hyperparameters regarding the used **environments**. We are using Gymnasium and the ALE, following the best practices by Machado et al. (2018) and the community.

| Hyperparameter | Value |
|---|---|
| ALE version | 0.8.1 |
| Gymnasium version | 0.29.1 |
| Environment version | v5 |
| Frameskip | 4 |
| Buffer Window Size | 4 |
| Observation Mode | RGB |
| Repeat Action Probability | 0.25 |
| Full Action Space | False |
| Continuous | False |

# E   Evaluation Metrics

We evaluate agent performance using three metrics following best practices from Agarwal et al. (2021): (1) **Human-Normalized Score (HNS)**, (2) **Interquartile Mean (IQM)**, and (3) **95% Confidence Intervals (CI)**.

**Human-Normalized Score (HNS).**   To compare performance across environments with varying reward scales, we use the HNS, defined via average agent score $A$, human score $H$, and random score $R$ (Badia et al., 2020):

$$\text{HNS} = \frac{A - R}{|H - R|}. \tag{5}$$

A value of 1.0 corresponds to human-level performance, while values near 0 suggest random behavior. Scores above 1.0 indicate superhuman performance.

**Aggregation via Interquartile Mean (IQM).**   To ensure robustness against outliers, we aggregate scores using the **IQM**, which computes the mean over the middle 50% of the data (the 25th to 75th percentiles). For all experiments, we report the IQM over 30 evaluation episodes (3 seeds × 10 episodes), aggregated across environments using the `rliable` library.

**Uncertainty and Significance Testing.**   We report **95% stratified bootstrap confidence intervals** to quantify uncertainty without assuming a parametric distribution. To assess statistical significance between OPC and the baselines, we employ:

- **Paired IQM CIs:** We compute the IQM over per-seed, per-environment differences to provide a robust summary of the performance shift.

- **Wilcoxon Signed-Rank Tests:** For per-environment analysis, we use this non-parametric test to determine if a variant significantly ($\alpha = 0.05$) outperforms the baseline.

**Object Detection Metrics.**   To assess the quality of YOLO and RT-DETR backends, we report three standard metrics:

- **F1 Score:** The harmonic mean of precision and recall, measuring object-identification stability and the avoidance of spurious predictions.

- **mAP@50:** Mean Average Precision at an Intersection-over-Union (IoU) threshold of 0.5, quantifying successful coarse localization.

- **mAP@50–95:** The average AP over IoU thresholds $\alpha \in \{0.50, 0.55, \ldots, 0.95\}$, penalizing both missed detections and imprecise bounding boxes:

$$\text{mAP@50–95} = \frac{1}{10} \sum_{\alpha=0.50}^{0.95} \text{AP}(\alpha). \tag{6}$$

**Craftax Reward Score**   The Craftax environment defines its own reward. Broken down, it gives the agent a positive reward between 1-8 for fulfilling achievements and penalizes losing health and similar. The reward is then normalized by the maximum possible number of achievements (226) and given as %$max$. For more information, see Matthews et al. (2024).

# F    Extended Experimental Results

To evaluate generalization, we measure how well agents trained in standard environments perform under visually and logically perturbed variants. These perturbations, drawn from the HackAtari benchmark, include visual changes (e.g., recolored objects, hidden enemies) and logical changes (e.g., modified object behavior or movement dynamics).

To provide further insight and give details, we include:

- **Table 7**: Full per-environment results including our baselines DQN, MDQN, Semantic Vector, and SCoBots. Additionally, it shows results for OPC with Rainbow as the backend, rather than PPO.

- **Section F.1**: Short ablation between pure object-centric Channels $\mathbf{C}_t$ and OPC ($\mathbf{X}_t$).

- **Section F.2**: We are testing our approach not only on Atari but also on Craftax with 3 Color Change Perturbation. More about this here.

- **Section F.3**: We include additional learning curves for PPO, object channels without pixel information, and OPC (similar to Figure 6b) as well as a scaling analysis of wall-clock runtime and memory allocation with respect to the number of channels.

- **Section F.4**. We include additional results for the synthetic noise experiments and YOLO-based detection pipelines discussed in Section 3.3, offering a more detailed breakdown of robustness under perceptual uncertainty.

- **Section F.5**: We further provide statistical significance analyses using the Wilcoxon signed-rank test to quantify performance differences between $\mathbf{C}_t$, $\mathbf{X}_t$, and PPO, as well as Rainbow.

- **Section F.6**: We evaluated whether the performance increase stemmed from the increased input size or the additional semantic information provided to the agent.

- **Section F.7**: We made a short ablation, evaluating the difference in robustness gain by using RAD (Laskin et al., 2020) compared to OPC.

Table 7: **Semantic channel representations improve visual robustness but struggle with logical perturbations.** This table reports the in-game episodic rewards (IQM + 95% CI) across 10 Atari environments, comparing OPC against several baselines. Agents are trained only on the original environment and evaluated zero-shot on perturbed variants (grey rows). OPC agents consistently outperform pixel and symbolic baselines under visual perturbations, demonstrating strong robustness to superficial changes. Best agent is highlighted in **bold**. When semantic channel representations outperform the DQN-like representation in the original game, the result is underlined.

| Game | PPO | | | Rainbow | | | Baselines | | | | |
|---|---|---|---|---|---|---|---|---|---|---|---|
| | DQN-like ($\mathbf{D}_t$) | Object Channels ($\mathbf{C}_t$) | OPC ($\mathbf{X}_t$) | DQN-like ($\mathbf{D}_t$) | Object Channels ($\mathbf{C}_t$) | OPC ($\mathbf{X}_t$) | DQN | MDQN | SemVec | SCoBots (NN) | SCoBots (DT) |
| Amidar | 551 [521,570] | 530 [513,542] | **907 [846,948]** | 381 [349,432] | 597 [451,768] | 658 [566,758] | 407 [319,541] | 723 [697,760] | 209 [184,232] | 180 [166,194] | 178 [160,197] |
| Player to Roller | 171 [137,207] | 524 [507,543] | **906 [845,945]** | 88 [70,106] | 652 [523,782] | 773 [634,885] | 80 [62,103] | 91 [72,113] | 209 [177,234] | 171 [160,185] | 178 [160,198] |
| Enemy to Pig | 554 [542,563] | 101 [84,117] | **898 [836,940]** | 645 [533,787] | 323 [248,432] | 738 [657,832] | 436 [337,573] | 740 [596,887] | 358 [88,753] | 74 [53,95] | 31 [31,31] |
| Bowling | 65 [64,67] | 62 [60,65] | 60 [60,60] | **82 [80,84]** | 69 [63,78] | 65 [61,74] | 28 [22,34] | 32 [30,34] | 62 [60,65] | — | — |
| Shift Player | **63 [61,66]** | 51 [31,64] | 60 [60,60] | 48 [20,76] | 58 [40,75] | 56 [49,60] | 30 [22,35] | 23 [18,28] | 62 [60,65] | — | — |
| Boxing | 95 [94,96] | 96 [94,98] | 96 [95,97] | 97 [96,98] | 91 [87,96] | **100 [99,100]** | 90 [86,93] | 97 [96,98] | 93 [90,96] | 96 [93,98] | 82 [75,87] |
| Boxers Red/Blue | 4 [1,8] | 98 [97,99] | 97 [95,98] | 1 [−1,5] | 92 [87,96] | **100 [99,100]** | −2 [−4,0] | −1 [−2,1] | 91 [89,93] | 95 [92,97] | 85 [79,89] |
| Breakout | 131 [84,183] | 280 [209,338] | **326 [273,360]** | 54 [37,85] | 86 [47,142] | 62 [32,144] | 56 [35,76] | 168 [121,224] | 37 [30,43] | — | — |
| Player/Ball Red | 7 [6,10] | **315 [261,358]** | 236 [174,296] | 7 [4,11] | 38 [21,99] | 151 [87,223] | 5 [4,6] | 27 [22,35] | 35 [27,42] | — | — |
| All Blocks Red | 156 [109,200] | 194 [130,255] | **305 [213,374]** | 28 [17,42] | 57 [31,109] | 124 [59,214] | 199 [156,244] | 304 [251,340] | 30 [23,37] | — | — |
| Freeway | 32 [31,33] | 33 [33,34] | 33 [33,34] | **34 [34,34]** | 33 [33,34] | **34 [34,34]** | 27 [16,33] | **34 [34,34]** | 31 [31,31] | 30 [29,31] | 27 [26,28] |
| Stop All Cars | 8 [0,20] | 22 [6,37] | 33 [33,34] | 0 [0,0] | 8 [0,20] | 34 [34,34] | 0 [0,0] | 33 [33,34] | 0 [0,0] | 33 [20,41] | **41 [41,41]** |
| All Cars Black | 24 [22,25] | 33 [33,34] | 33 [33,34] | 25 [24,26] | 33 [32,33] | **34 [34,34]** | 12 [9,14] | 25 [25,26] | 31 [31,31] | 30 [29,31] | 27 [26,28] |
| Frostbite | 300 [292,306] | 282 [275,290] | 298 [289,307] | 2749 [2374,3140] | 3254 [2869,3632] | 2461 [2208,2792] | 3445 [3012,3766] | **4349 [2554,5791]** | 265 [258,390] | — | — |
| Static Ice | 40 [40,40] | 25 [0,72] | 1 [0,6] | **684 [214,1445]** | 40 [14,81] | 180 [136,428] | 42 [22,71] | 26 [7,64] | 30 [0,82] | — | — |
| MsPacman | 3001 [2803,3396] | 6130 [5603,6681] | 6305 [5523,7383] | 3004 [2701,3552] | 3854 [3592,4167] | 4459 [3691,5167] | 2469 [2326,2614] | 2406 [2269,2560] | 1919 [1406,2631] | — | — |
| 2nd Level | 97 [60,184] | 334 [269,407] | 87 [47,132] | **1079 [911,1271]** | 698 [626,775] | 1000 [895,1137] | 469 [376,581] | 384 [329,461] | 72 [60,80] | — | — |
| Pong | 15 [14,16] | 19 [18,19] | 19 [19,19] | 18 [17,19] | **20 [20,20]** | **20 [19,20]** | 18 [17,19] | 17 [16,18] | 19 [18,20] | 18 [16,19] | 16 [14,17] |
| Lazy Opponent | −8 [−11,−6] | **18 [17,18]** | −4 [−13,6] | −4 [−8,0] | 16 [15,17] | −4 [−12,4] | −2 [−4,0] | −5 [−8,−1] | −19 [−21,−16] | −21 [−21,−20] | −11 [−13,−8] |
| Hidden Opponent | −18 [−19,−17] | 18 [18,19] | 18 [17,18] | −21 [−21,−20] | **19 [19,20]** | 19 [18,20] | −21 [−21,−20] | −19 [−20,−18] | −21 [−21,−21] | −20 [−21,−20] | −13 [−16,−9] |
| Riverraid | 7668 [7279,8046] | 8020 [7824,8216] | **9786 [9594,9992]** | 3009 [2203,4161] | 3043 [2714,3396] | 2974 [2281,3902] | 9202 [8895,9649] | 8633 [8201,9055] | 3306 [3069,3567] | 2436 [2333,2576] | 2441 [2321,2563] |
| Color Change | 441 [262,772] | 8263 [8026,8504] | **9644 [9419,9820]** | 509 [432,600] | 3546 [3118,4086] | 2622 [1953,3833] | 285 [210,391] | 868 [754,1034] | 3525 [3256,3786] | 1601 [1517,1703] | 1428 [1358,1502] |
| Linear River | 6932 [5770,7722] | 1969 [1574,2406] | 9634 [9371,9868] | 5018 [3754,6349] | 3282 [3191,3375] | 3593 [2605,4611] | 10901 [9654,12311] | **11568 [10666,12444]** | 356 [301,388] | 3168 [2698,3518] | 1551 [1414,1736] |
| SpaceInvaders | 744 [703,796] | 1556 [1267,1892] | 1676 [1287,2081] | 1003 [886,1185] | 972 [823,1174] | 1510 [1325,1701] | 1248 [1096,1389] | **1759 [1473,2078]** | 358 [308,419] | 343 [306,380] | 389 [335,449] |
| Shields off by 3 | 621 [558,683] | 1429 [1061,1789] | **1782 [1496,2079]** | 710 [571,899] | 987 [840,1164] | 1728 [1524,1940] | 564 [478,670] | 729 [538,927] | 415 [342,485] | — | — |

## F.1 Ablation Study: Object Channels vs. OPC

To isolate the contribution of raw visual context versus semantic abstraction, we analyze the performance delta between the purely object-centric representation $\mathbf{C}_t$ (object channels) and the hybrid representation $\mathbf{X}_t$ (OPC). While $\mathbf{C}_t$ provides a theoretically "clean" state that is invariant to many visual perturbations, it suffers from a lack of *perceptual grounding* in environments where task-relevant information exists outside of the predefined object classes.

**The Cost of Excessive Abstraction.** The primary failure mode of $\mathbf{C}_t$ is observed in environments like *Riverraid*. In this domain, the agent must navigate within the narrow, jagged boundaries of a river. While the object set $\mathbb{O}_t$ captures the player, enemies, and fuel tanks, the river's non-linear coastline is typically not modeled as a discrete object class in standard Atari object-detection schemas (e.g., OCAtari).

Consequently, an agent trained only on $\mathbf{C}_t$ is effectively "blind" to the terrain. Without the pixel channel to provide geometric context for the riverbanks, the agent fails to learn the steering logic needed to avoid collisions with the environment. This results in significantly lower scores than pixel-based baselines, despite the agent having perfect information about enemy positions.

The hybrid representation $\mathbf{X}_t = [\mathbf{D}_t; \mathbf{C}_t]$ resolves this by using the greyscale pixel channel $\mathbf{D}_t$ as a geometric prior. With OPC, the agent uses the object channels to identify high-level entities (task logic) while simultaneously grounding them in the static environment (geometry) via the pixel channel. As shown in our results, OPC maintains the robustness of symbolic logic without sacrificing the fine-grained spatial awareness required for navigation.

**Formal Informational Comparison.** From an information-theoretic perspective, $\mathbf{C}_t$ is a filtered subset of the input $\mathbf{I}_t$ designed to maximize the signal-to-noise ratio for task-relevant objects. However, the mapping $f : \mathbf{I}_t \to \mathbf{C}_t$ in itself can lose information. $\mathbf{X}_t$ functions as a residual representation:

$$\mathbf{X}_t = \mathbf{D}_t \oplus \text{Semantic Entities}(\mathbf{I}_t), \tag{7}$$

where $\mathbf{D}_t$ ensures that any features $\phi \in \mathbf{I}_t$ that were not captured by the object detector, yet are critical for reward maximization, remain available to the policy. This suggests that for general-purpose DRL, the bridge between high-level semantics and low-level visual evidence is a necessary architectural requirement for robust intelligence.

**Quantifying the Information Gap.** To understand the necessity of the hybrid bridge, we compare the purely symbolic spatial encoding ($\mathbf{C}_t$) against the hybrid representation ($\mathbf{X}_t$). This comparison highlights a fundamental trade-off in representation design:

- **Invariance vs. Completeness:** Object channels ($\mathbf{C}_t$) offer maximum invariance to visual distribution shifts by entirely discarding pixel data. However, as seen in *Riverraid*, this leads to a 65% performance drop compared to OPC because the agent loses sight of non-objectified environmental constraints (the river coastline).

- **The Grounding Requirement:** OPC ($\mathbf{X}_t$) serves as a residual learner. The pixel channel acts as a safety net, capturing any geometric features that the object detector misses. However, keeping the pixel channel in the input representation still allows the agents to learn shortcuts. While the overall performance is better, they are not as robust as pure object channels.

This result suggests that for deep agents to achieve human-like intelligence, they must not only identify *what* objects are present (semantics) but also *where* those objects exist in relation to the raw geometric context (grounding).

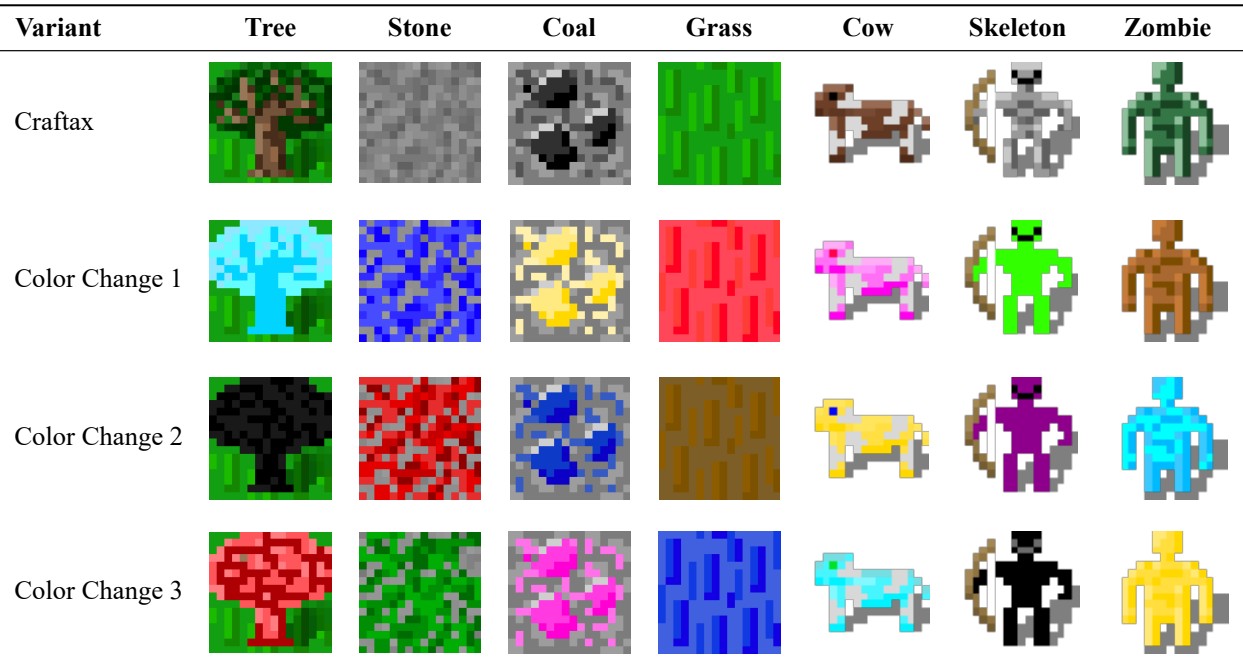

Figure 7: Color Changes in Craftax. Each line corresponds to one color change modification in Table 8.

## F.2 Experiments on Craftax

While the Arcade Learning Environment (ALE) or Atari remains the standard for reactive control and perception, its near-deterministic nature often allows agents to succeed through pattern memorization rather than true generalization. This is one reason why we use ALE as our primary testbed. In contrast, Craftax, a JAX-native environment introduced at ICML 2024 (Matthews et al., 2024), utilizes procedural generation and complex survival mechanics to mandate hierarchical reasoning and long-term planning. Craftax offers a significant "efficiency-complexity" advantage, as its GPU-accelerated framework allows a full billion-step training run to complete in under an hour on a single machine, a feat previously impossible with CPU-bound benchmarks like Atari. Furthermore, Craftax needs a different type of planning, with very long reward horizons and complex task structures, making it a good addition to our Atari testbed.

Hyperparameters and experimental setup are taken from Matthews et al. (2024). The classical input representation here differs from the classical ALE one. As such, we do not use frame-skipping or $84 \times 84$ image stacks, but instead use the setup of the original work to facilitate comparisons. All channels in OPC are adapted accordingly. The metric is the percentage of achievements fulfilled and described in Section E.

Table 8: **In complex games like Craftax, semantic channel representations improve robustness.** Craftax results after 10M training steps, showing the IQM+95%CI over 3 seeds. The %*max* metric from Matthews et al. (2024) is shown.

| Game | PPO | OPC ($\mathbf{X}_t$) | Object Channels ($\mathbf{C}_t$) |
| --- | --- | --- | --- |
| Craftax | **6.15** [5.41,6.85] | 5.52 [4.66,6.13] | 4.99 [4.69,5.30] |
| Color Change 1 | 0.18 [0.04,0.32] | **5.85** [4.25,7.10] | 5.11 [4.86,5.38] |
| Color Change 2 | 0.60 [0.38,0.85] | 3.96 [3.28,4.91] | **4.83** [4.55,5.08] |
| Color Change 3 | 0.57 [0.24,0.82] | 1.37 [0.77,2.04] | **4.97** [4.66,5.22] |

### F.3  Scalability

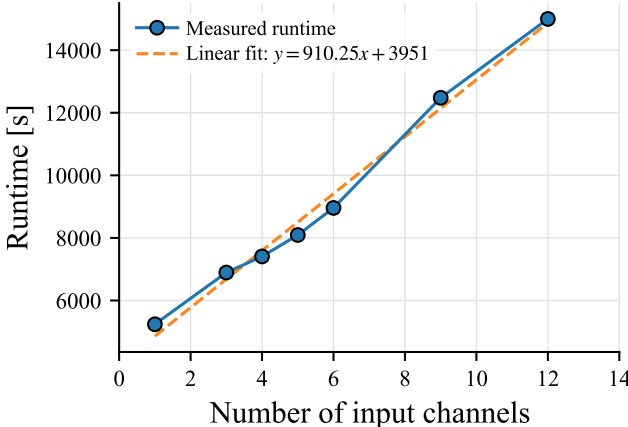

Figure 8: **Runtime increases proportionally with channel dimensionality.** Wall-clock training time scales approximately linearly with the number of channels, reflecting the additional convolutional cost introduced by expanded input depth.

**GPU Memory Footprint.**  Beyond wall-clock time, OPC increases GPU memory consumption proportionally to $|\mathcal{C}|$, as the input tensor grows from $(1 \times 84 \times 84)$ to $((|\mathcal{C}| + 1) \times 84 \times 84)$. Table 9 reports peak reserved and allocated GPU memory for PPO and OPC across all evaluated games. For on-policy PPO, reserved memory remains constant at $0.537$ GB across all games, as the input size is game-independent. For OPC, memory scales approximately linearly with $|\mathcal{C}|$, similar to the runtime (cf. Figure 8). For off-policy Rainbow, the replay buffer is the dominant contributor to memory usage.

Table 9: **Memory scales linear with the number of channels.** Peak reserved and allocated GPU memory (in GB) for PPO and OPC (PPO) across all evaluated games. $|\mathcal{C}|$ denotes the number of object categories. All experiments were conducted on NVIDIA V100 32 GB GPUs.

| Game | $|\mathcal{C}|$ | PPO Alloc. (GB) | PPO Reserv. (GB) | OPC (PPO) Alloc. (GB) | OPC (PPO) Reserv. (GB) | Ratio |
|---|---|---|---|---|---|---|
| Freeway | 3 | 0.241 | 0.537 | 0.678 | 1.132 | 2.11× |
| Boxing | 5 | 0.241 | 0.537 | 0.969 | 1.642 | 3.06× |
| Pong | 5 | 0.241 | 0.537 | 0.969 | 1.642 | 3.06× |
| Breakout | 6 | 0.241 | 0.537 | 1.115 | 1.914 | 3.56× |
| Amidar | 7 | 0.241 | 0.537 | 1.206 | 2.166 | 4.03× |
| MsPacman | 7 | 0.241 | 0.537 | 1.206 | 2.166 | 4.03× |
| SpaceInvaders | 8 | 0.241 | 0.537 | 1.406 | 2.424 | 4.51× |
| Riverraid | 10 | 0.241 | 0.537 | 1.698 | 2.917 | 5.43× |
| Frostbite | 12 | 0.241 | 0.537 | 1.989 | 3.347 | 6.23× |

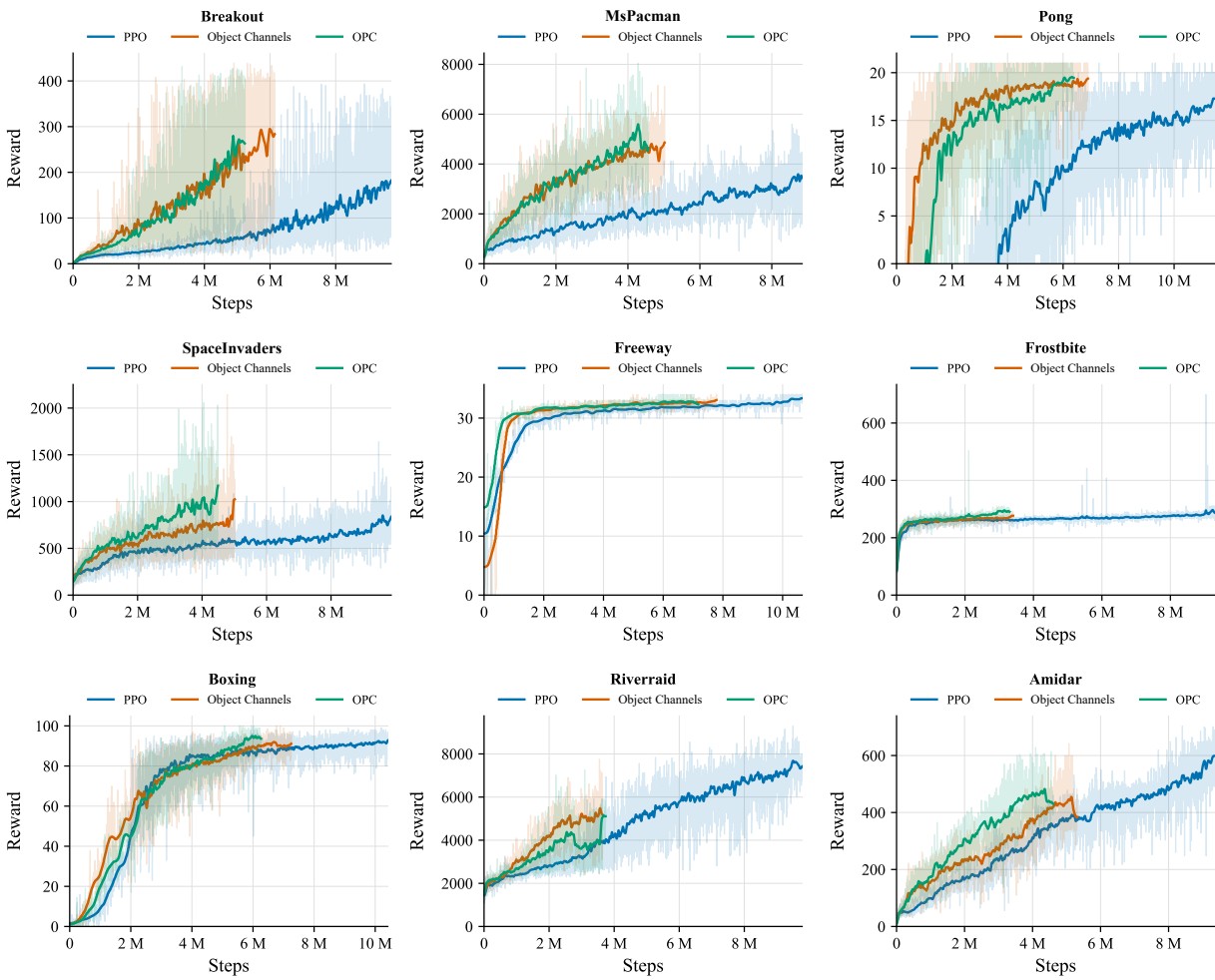

Figure 9: **Despite their higher computational cost, agents using semantic channel representations ($\mathsf{C}_t$ and $\mathsf{X}_t$) outperform standard PPO in most settings within a fixed 90-minute training budget.** Final performance results are shown in Figure 6. Shaded regions indicate variance across 3 training seeds.

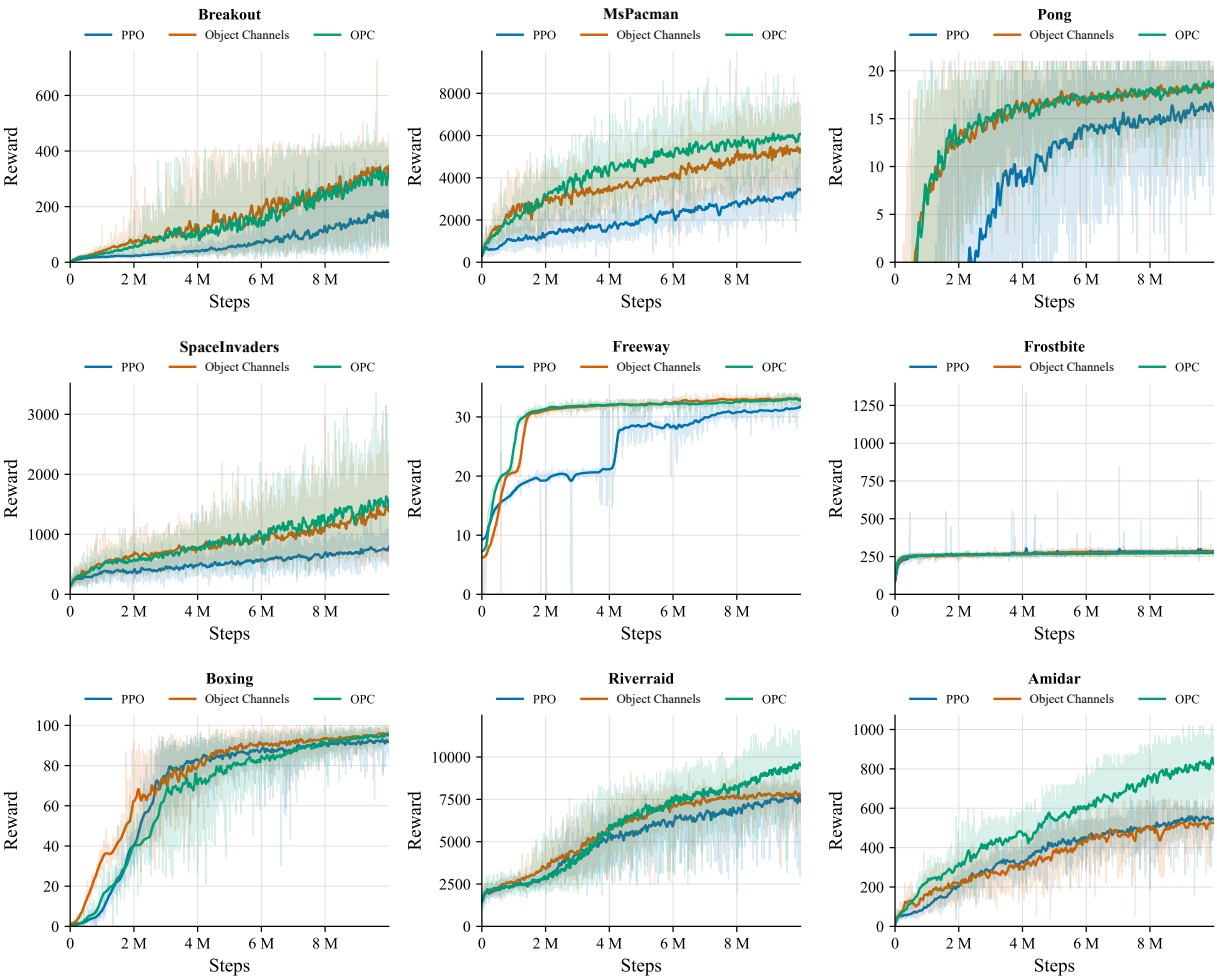

Figure 10: **Sample Efficiency and Training Convergence.** Reward curves are plotted against training progress (10M steps / 40M frames) across nine environments. The results demonstrate that semantic channel representations, especially OPC, significantly improve sample efficiency and achieve higher asymptotic rewards compared to the baseline PPO. Shaded regions indicate variance across 3 training seeds.

### F.4 Extended Noise Results

Table 10: **Zero-shot Generalization under Perception Noise.** Comparison of IQM episodic rewards between pure object channels ($\mathbf{C}_t$) without pixel information and **OPC** ($\mathbf{X}_t$) under synthetic noise. Results demonstrate the robustness of hybrid representations to distribution shifts between training and test-time noise levels.

| Game (Train/Test Noise) | Object Channels ($\mathbf{C}_t$) | | | OPC ($\mathbf{X}_t$) | | |
|---|---|---|---|---|---|---|
| | **2%/2%** | **2%/5%** | **2%/10%** | **2%/2%** | **2%/5%** | **2%/10%** |
| **Pong** | $18_{[17,19]}$ | $18_{[17,19]}$ | $17_{[16,18]}$ | $19_{[18,19]}$ | $19_{[18,19]}$ | $19_{[18,20]}$ |
| **Freeway** | $32_{[31,32]}$ | $32_{[31,33]}$ | $31_{[30,32]}$ | $33_{[33,33]}$ | $33_{[32,33]}$ | $33_{[32,33]}$ |
| **Breakout** | $224_{[150,296]}$ | $103_{[73,173]}$ | $61_{[49,77]}$ | $278_{[244,312]}$ | $225_{[171,284]}$ | $118_{[96,164]}$ |
| **Game (Train/Test Noise)** | **5%/5%** | **5%/10%** | **10%/10%** | **5%/5%** | **5%/10%** | **10%/10%** |
| **Pong** | $19_{[19,20]}$ | $18_{[18,19]}$ | $17_{[16,18]}$ | $18_{[16,18]}$ | $18_{[16,19]}$ | $18_{[17,19]}$ |
| **Freeway** | $33_{[33,24]}$ | $33_{[33,33]}$ | $33_{[32,33]}$ | $32_{[30,33]}$ | $31_{[29,33]}$ | $33_{[32,33]}$ |
| **Breakout** | $125_{[97,184]}$ | $69_{[57,83]}$ | $78_{[67,89]}$ | $256_{[187,310]}$ | $152_{[121,201]}$ | $227_{[180,281]}$ |

**Pong**: Object Channels ($\mathbf{C}_t$)

| Train Fail Prob. | Test Fail Probability | | | |
|---|---|---|---|---|
| | 0% | 2% | 5% | 10% |
| 2% | $19_{[19,20]}$ | $18_{[17,19]}$ | $18_{[17,19]}$ | $17_{[16,18]}$ |
| 5% | $19_{[18,20]}$ | $20_{[19,20]}$ | $19_{[19,20]}$ | $18_{[17,19]}$ |
| 10% | $18_{[16,18]}$ | $17_{[16,18]}$ | $17_{[16,18]}$ | $17_{[16,18]}$ |

**Pong**: OPC ($\mathbf{X}_t$)

| Train Fail Prob. | Test Fail Probability | | | |
|---|---|---|---|---|
| | 0% | 2% | 5% | 10% |
| 2% | $19_{[19,20]}$ | $19_{[18,19]}$ | $19_{[18,19]}$ | $19_{[18,19]}$ |
| 5% | $18_{[17,19]}$ | $18_{[17,19]}$ | $18_{[16,18]}$ | $18_{[16,19]}$ |
| 10% | $19_{[18,20]}$ | $19_{[18,19]}$ | $19_{[18,19]}$ | $18_{[17,19]}$ |

**Breakout**: Object Channels ($\mathbf{C}_t$)

| Train Fail Prob. | Test Fail Probability | | | |
|---|---|---|---|---|
| | 0% | 2% | 5% | 10% |
| 2% | $165_{[97,246]}$ | $224_{[149,296]}$ | $103_{[73,174]}$ | $61_{[49,77]}$ |
| 5% | $84_{[49,163]}$ | $149_{[102,216]}$ | $125_{[97,184]}$ | $69_{[57,83]}$ |
| 10% | $47_{[31,68]}$ | $98_{[78,127]}$ | $81_{[74,91]}$ | $78_{[67,89]}$ |

**Breakout**: OPC ($\mathbf{X}_t$)

| Train Fail Prob. | Test Fail Probability | | | |
|---|---|---|---|---|
| | 0% | 2% | 5% | 10% |
| 2% | $262_{[163,329]}$ | $278_{[244,311]}$ | $225_{[171,285]}$ | $118_{[96,164]}$ |
| 5% | $150_{[63,239]}$ | $221_{[161,283]}$ | $256_{[186,310]}$ | $152_{[120,200]}$ |
| 10% | $94_{[68,131]}$ | $164_{[125,224]}$ | $199_{[141,259]}$ | $227_{[180,283]}$ |

**Freeway**: Object Channels ($\mathbf{C}_t$)

| Train Fail Prob. | Test Fail Probability | | | |
|---|---|---|---|---|
| | 0% | 2% | 5% | 10% |
| 2% | $32_{[32,32]}$ | $32_{[31,32]}$ | $32_{[31,33]}$ | $31_{[30,32]}$ |
| 5% | $34_{[33,34]}$ | $33_{[33,33]}$ | $33_{[33,34]}$ | $33_{[33,33]}$ |
| 10% | $33_{[33,33]}$ | $33_{[33,33]}$ | $33_{[32,33]}$ | $33_{[32,33]}$ |

**Freeway**: OPC ($\mathbf{X}_t$)

| Train Fail Prob. | Test Fail Probability | | | |
|---|---|---|---|---|
| | 0% | 2% | 5% | 10% |
| 2% | $33_{[33,33]}$ | $33_{[33,33]}$ | $33_{[32,33]}$ | $33_{[32,33]}$ |
| 5% | $32_{[30,33]}$ | $32_{[30,33]}$ | $32_{[30,30]}$ | $31_{[29,33]}$ |
| 10% | $32_{[32,33]}$ | $32_{[32,33]}$ | $33_{[32,33]}$ | $33_{[32,33]}$ |

Figure 11: **Robustness to Imperfect Object Detection.** Performance (IQM) across three environments under varying levels of object drop probability. We compare pure object channels without pixel information ($\mathbf{C}_t$) (left) against **OPC** ($\mathbf{X}_t$) (right) to demonstrate how pixel-grounding provides a visual safety net during detector failure.

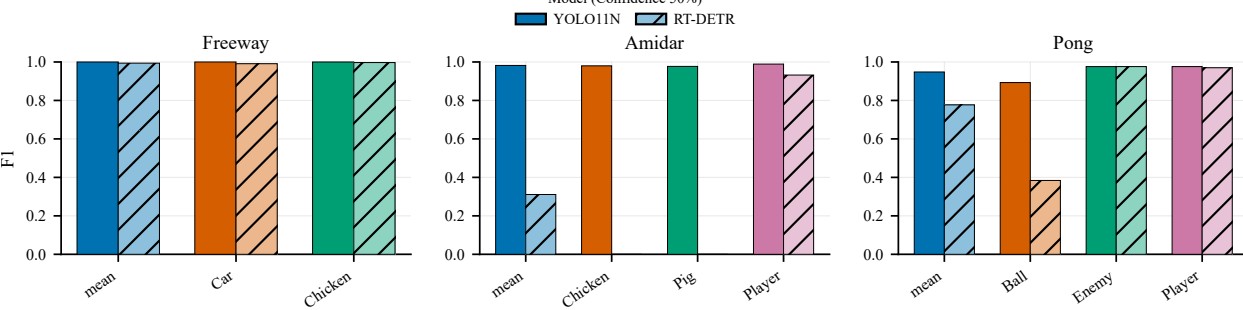

Figure 12: **Quality of Object Detection and Learning Curves for YOLO and RT-DETR.** Illustrating the difference in stability and quality between different detectors and objects. YOLO outperforms RT-DETR in all three Atari games. While both achieve high confidence in Freeway, both approaches struggle in Amidar and Pong.

Figure 13: **Object detection robustness and performance across Atari environments.** Detection performance at a fixed confidence threshold ($\tau = 0.50$) highlights the variance in object visibility and class complexity across games.

Table 11: **Modern object detectors can reliably substitute perfect object annotations when detection quality is adequate.** Across several games, YOLO-based detectors closely match the OCAtari REM baseline, indicating that oracle-based object extraction can be replaced. Performance collapses only when critical objects are consistently missed, emphasizing that robustness depends on preserving task-relevant structure. Grey values indicate a collapse larger than an order of magnitude.

| Detector | Object Channels ($\mathbf{C}_t$) | | | OPC ($\mathbf{X}_t$) | | |
|---|---|---|---|---|---|---|
| | Freeway | Amidar | Pong | Freeway | Amidar | Pong |
| OCAtari REM | $34_{[33,34]}$ | $543_{[532,562]}$ | $19_{[18,20]}$ | $33_{[33,33]}$ | $903_{[840,945]}$ | $19_{[19,19]}$ |
| OCAtari VEM | $34_{[33,34]}$ | — | $19_{[17,20]}$ | $33_{[33,33]}$ | — | $20_{[19,20]}$ |
| YOLO11n | $34_{[34,34]}$ | $562_{[533,601]}$ | $14_{[10,16]}$ | $33_{[32,33]}$ | $702_{[598,771]}$ | $18_{[18,19]}$ |
| YOLO11l | $34_{[33,34]}$ | $593.2_{[564,594]}$ | $-3_{[-11,5]}\downarrow$ | $32_{[32,33]}$ | $639_{[541,746]}$ | $18_{[16,18]}$ |
| YOLO12n | $34_{[34,34]}$ | $573.4_{[500,593]}$ | $-3_{[-11,5]}\downarrow$ | $33_{[32,33]}$ | $653_{[586,727]}$ | $18_{[18,19]}$ |
| YOLO12l | $34_{[34,34]}$ | $632_{[592,643]}$ | $-12_{[-14,-10]}\downarrow$ | $32_{[32,33]}$ | $578_{[492,647]}$ | $16_{[13,18]}$ |
| RT-DETR | $9_{[7,10]}$ | $1_{[0,6]}\downarrow$ | $-8_{[-15,2]}\downarrow$ | $15_{[14,15]}$ | $46_{[33,59]}\downarrow$ | $16_{[14,17]}$ |

### F.5 Statistical Significance

To evaluate the statistical significance of OPC relative to the DQN-like pixel baseline ($\mathbf{C}_t$), we conducted Wilcoxon signed-rank tests ($n = 3$ seeds per agent) independently for each environment. We performed one-sided tests in both directions—testing whether an agent significantly outperforms or is outperformed by the baseline. Bold $p$-values in **Table 12** indicate statistical significance ($\alpha = 0.05$) for PPO and Rainbow backbones, respectively.

The bidirectional tests reveal consistent improvements of OPC and object channels compared to their baselines. The improvements are more prominent in PPO. In games such as *Breakout* and *Freeway*, it achieves $p$-values smaller than 0.1%, reflecting robust performance gains across seeds.

To complement the per-environment tests, we additionally computed paired IQM differences across all environments, using a stratified bootstrap to estimate 95% confidence intervals. For the comparison between PPO and OPC, we observe an average improvement of $0.0654$ $[0.0464, 0.0855]$ (measured by the agents' HNS scores). For Rainbow, the shift is $0.0708$ $[0.0252, 0.0996]$. These results summarize the aggregate performance shift for the semantic channel representations relative to the baseline. Consistent with the Wilcoxon significance analysis, OPC exhibit a positive IQM shift.

Table 12: **Representation Robustness Comparison.** Two-sided $p$-values ($t$) comparing semantic channels and OPC against pixel-based baselines. The summary count demonstrates that OPC provides the most consistent path to robustness, frequently outperforming or matching the baseline where pure abstraction (object channels $\mathbf{C}_t$) fails.

| | PPO | | Rainbow | |
|---|---|---|---|---|
| **Environment** | $\mathbf{C}_t$ | OPC ($\mathbf{X}_t$) | $\mathbf{C}_t$ | OPC ($\mathbf{X}_t$) |
| Amidar | **0.004** ↓ | **0.002** ↑ | **0.002** ↑ | **0.002** ↑ |
| Bowling | **0.002** ↓ | **0.002** ↓ | **0.002** ↓ | **0.037** ↓ |
| Boxing | 0.652 | 0.084 | **0.002** ↓ | 0.113 |
| Breakout | 0.768 | **0.027** ↑ | 0.432 | 1.000 |
| Freeway | **0.008** ↑ | **0.031** ↑ | 1.000 | 0.625 |
| Frostbite | **0.002** ↓ | **0.016** ↓ | **0.049** ↑ | 0.232 |
| MsPacman | **0.002** ↑ | **0.002** ↑ | 0.084 | **0.006** ↑ |
| Pong | **0.004** ↑ | **0.002** ↑ | 0.109 | 0.086 |
| Riverraid | **0.002** ↑ | **0.006** ↑ | 0.193 | **0.014** ↑ |
| SpaceInvaders | **0.004** ↑ | **0.002** ↑ | — | — |
| **Sig. Wins** (↑) | 5 | 7 | 2 | 3 |
| **Sig. Losses** (↓) | 3 | 2 | 2 | 1 |

### F.6 Improvement by Capacity or Semantics?

A concern about hybrid representations such as OPC is that the observed improvements may simply come from feeding the agent a larger input tensor: OPC stacks $|\mathbb{C}|+1$ channels, whereas the pixel baseline $\mathbf{D}_t$ consists of a single grayscale channel. One might therefore ask whether the extra convolutional capacity at the first layer is responsible for the gains.

To isolate the contribution of the *semantic* signal, we did a small ablation, training an extended PPO agent (**Ex-PPO**): a PPO agent on the same pixel input $\mathbf{D}_t$, but with input and first-layer width scaled to match the OPC tensor. For this, we copied the first channel ($\mathbf{D}_t$) $n$ times to match the number of channels of OPC, depending on the game. The only difference between Ex-PPO and OPC is therefore the *content* of the additional channels.

Table 13 reports this comparison on four games with their respective HackAtari variants used throughout the paper. In the default environments, Ex-PPO performs on par with, or only modestly above, the standard PPO baseline ($\mathbf{D}_t$) and remains inferior to OPC. Under zero-shot perturbations, the picture is even clearer: Ex-PPO collapses in the same way as PPO whenever the perturbation attacks a task-relevant semantic cue. Taken together, these results support our central claim: the advantage of OPC comes from the *semantic information* injected into the representation, not from the increased capacity at the perception stage.

Table 13: **Capacity-matched ablation: PPO vs. OPC vs. Ex-PPO.** In-game episodic rewards (IQM with 95 % stratified bootstrap CI) on the default environments and their HackAtari variants. Agents are trained only on the unperturbed environment and evaluated zero-shot on the perturbed variants. **Bold** marks the best entry per row. PPO denotes the DQN-like pixel baseline ($\mathbf{D}_t$; taken from Table 7). Ex-PPO uses the same pixel input but repeated channels to match the channel count of OPC, so that Ex-PPO and OPC differ *only* in whether the extra channels carry semantic content. OPC consistently dominates whenever the task depends on the semantic channels, while Ex-PPO tracks the pixel baseline. This is evidence that the gain stems from semantics, not from capacity.

| Game | PPO backbone | | |
| --- | --- | --- | --- |
| | PPO ($\mathbf{D}_t$) | OPC ($\mathbf{X}_t$) | Ex-PPO |
| *Amidar* | 551 [521, 570] | **907** [**846**, **948**] | 503 [428, 558] |
| Player to Roller | 171 [137, 207] | **906** [**845**, **945**] | 137 [ 92, 186] |
| Enemy to Pig | 554 [542, 563] | **898** [**836**, **940**] | 375 [319, 444] |
| *MsPacman* | 3001 [2803, 3396] | **6305** [**5523**, **7383**] | 4188 [3321, 4774] |
| 2nd Level | 97 [60, 184] | 87 [47, 132] | **364** [**246**, **450**] |
| *Pong* | 15 [14, 16] | **19** [**19**, **19**] | 18 [17, 19] |
| Lazy Opponent | $-8$ [$-11, -6$] | **$-4$** [**$-13$**, **6**] | $-17$ [$-19, -14$] |
| Hidden Opponent | $-18$ [$-19, -17$] | 18 [17, 18] | $-10$ [$-18, 1$] |
| *Riverraid* | 7668 [7279, 8046] | **9786** [**9594**, **9992**] | 7166 [7000, 7320] |
| Color Change | 441 [ 262, 772] | **9644** [**9419**, **9820**] | 244 [ 173, 345] |
| Linear River | 6932 [5770, 7722] | **9634** [**9371**, **9868**] | 4475 [3667, 4819] |

### F.7 Semantic Channels vs. Pixel-Space Data Augmentation

The previous ablation showed that the gains of OPC are not a pure capacity effect. A complementary question is whether our improved robustness can be recovered *without* object information, by using a strong pixel-space regularizer. Reinforcement Learning with Augmented Data (Laskin et al., 2020) (RAD) is the canonical baseline for this: a pixel-only PPO agent trained with random crop and color-jitter augmentations that directly attack the visual shortcuts exploited by standard CNN policies. If the robustness gap between PPO and OPC stems from brittle pixel-level features, one would expect data augmentation to close most of that gap on visual perturbations.

To test this, we trained **PPO+RAD** in four environments, using random translation and color jitter[3], and evaluate zero-shot on the HackAtari variants. Table 14 reports the comparison. The results show that RAD can improve robustness, e.g., in *Player to Roller Amidar* or *Hidden Opponent Pong*, but struggles with others. Overall, it shows a good extension of our work. A study extending our short ablation to see the full extent of RAD with HackAtari modifications or even combining both paradigms, RAD and OPC, is beyond the current scope, but it is a promising avenue for future investigation.

Table 14: **Semantic channels vs. pixel-space data augmentation: PPO vs. OPC vs. PPO+RAD.** In-game episodic rewards (IQM with 95 % stratified bootstrap CI) on four environments and their HackAtari variants. All agents are trained only on the default environment and evaluated zero-shot on the perturbed variants. **Bold** marks the best entry per row. PPO denotes the DQN-like pixel baseline ($\mathbf{D}_t$; taken from Table 7). PPO+RAD is the same pixel backbone trained with random translation, cropping, and color jitter[3], following Laskin et al. (2020). OPC shows better improvements in variants that attack semantic content (color, class, or removed entity), while PPO+RAD shows great results, especially in Riverraid. This is consistent with the view that structural representational choices and data-driven augmentations are complementary.

| | PPO backbone | | |
| Game | PPO ($\mathbf{D}_t$) | OPC ($\mathbf{X}_t$) | PPO+RAD |
|---|---|---|---|
| *Amidar* | 551 [ 521, 570] | **907** [ **846**, **948**] | 507 [ 469, 538] |
| Player to Roller | 171 [ 137, 207] | **906** [ **845**, **945**] | 462 [ 427, 505] |
| Enemy to Pig | 554 [ 542, 563] | **898** [ **836**, **940**] | 471 [ 441, 496] |
| *MsPacman* | 3001 [2803, 3396] | **6305** [**5523**, **7383**] | 2885 [2472, 3096] |
| 2nd Level | 97 [ 60, 184] | 87 [ 47, 132] | **404** [**184**, **903**] |
| *Pong* | 15 [14, 16] | **19** [**19**, **19**] | 18 [17, 19] |
| Lazy Opponent | $-8$ [$-11$, $-6$] | $-4$ [$-13$, 6] | **4** [$-$**6**, **11**] |
| Hidden Opponent | $-18$ [$-19$, $-17$] | **18** [**17**, **18**] | 13 [10, 16] |
| *Riverraid* | 7668 [ 7279, 8046] | 9786 [ 9594, 9992] | **10499** [**10323**, **10738**] |
| Color Change | 441 [ 262, 772] | **9644** [**9419**, **9820**] | 624 [ 469, 789] |
| Linear River | 6932 [ 5770, 7722] | 9634 [ 9371, 9868] | **11425** [ **9967**, **12651**] |

---

[3]Standard color jitter requires RGB input; however, due to codebase limitations, we implemented a grayscale equivalent. We simulate stochastic variance in luminance and contrast by applying a random exponential transformation $x^{2\gamma}$, where $x$ represents the normalized pixel intensity of the grayscale image and the exponent $\gamma$ is sampled from Beta(2, 2). This centered distribution allows for a symmetric range of non-linear "gamma" shifts, inducing distributional variance analogous to traditional color-space augmentations.

## G    Object Extraction with OCAtari

To extract structured object representations from Atari environments, we leverage **OCAtari** Delfosse et al. (2024b), a lightweight framework built on top of the ALE. In this work, we exclusively use OCAtari's **RAM Extraction Method (REM)**, which decodes semantically meaningful object properties directly from the emulator's memory. These object states serve as input to our masking module, which generates structured, task-agnostic representations.

**Installation.**    OCAtari and its dependencies can be installed using:

```
pip install ocatari
pip install gymnasium[atari]
```

**Object Information.**    OCAtari's REM provides access to object-level states (e.g., category, position, size) for each supported game. These mappings are hand-designed for each game and remain consistent across runs, enabling reliable, interpretable abstraction. A minimal usage example is shown in Listing 1.

Each object is returned as a structured entity with (at least):

- `category` (e.g., "Ball", "Player", "Score"),

- `x,y`: pixel coordinates relative to the upper left corner,

- `w,h`: width and height in pixels,

- `dx,dy`: the difference in x and y between this frame and the last.

Head-up display (HUD) elements (e.g., score counters) are optional and were not included in this work. Most HUD elements would be rendered irrelevant or unhelpful when incorporated into the masking approaches; as such, they would add noise rather than help the agent in their decision. However, in some games, HUD elements can play a major role in decision-making, such as in Seaquest or BankHeist. In such cases, it can be helpful to use HUDs and incorporate them as additional objects or channels in the representation. To do so, change the OCATari parameter `hud` to True.

**Usage.**    OCAtari can be used like any other GymWrapper. An example is shown in Listing 1.

```python
from ocatari.core import OCAtari

env = OCAtari("Pong", mode="ram")
action_space = env.action_space
obs, info = env.reset()

for _ in range(1000):
    #obs, reward, terminated, truncated, info
    s_tuple = env.step(action_space.sample())
    terminated, truncated = s_tuple[2:4]

    for obj in env.objects:
        print(obj.category, obj.xy, obj.wh)
    if terminated or truncated:
        break
```

Listing 1: Using OCAtari with REM to extract object properties in `Pong`.

**Resources.**    For a complete list of supported environments and advanced usage, visit `https://oc-atari.readthedocs.io`.

# H   Atari Game Variants

Below, we briefly describe each Atari game variant used in our study. Descriptions are done by us or taken from the ALE Documentation[4]. These variants are visualized in Figure 14 and created using the HackAtari Environment (Delfosse et al., 2024a). The modification list to create each variant is given in Table 15. This can be used to visualize or evaluate the models' performances.

To create any of the following variants for a specific Atari game (`env_id`), you can simply pass the modification list (`modifs`) for the variant into HackAtari:

```
env = HackAtari(env_id, modifs=modifs)
```

## H.1   Amidar Variants

**Amidar:** *Amidar* is similar to Pac-Man: You are trying to visit all places on a 2-dimensional grid while simultaneously avoiding your enemies.

**Enemy to Pig:** Change the enemy warriors into Pigs. The game logic stays the same.

**Player to Roller:** Changing the sprite of the player figure from a human to a paint roller. The game logic stays the same.

## H.2   Bowling Variants

**Bowling:** Your goal is to score as many points as possible in the game of *Bowling*. A game consists of 10 frames, and you have two tries per frame.

**Shift Player:** The player starts closer to the pins.

## H.3   Boxing Variants

**Boxing:** The standard *Boxing* environment where two players compete to land punches.

**Boxers Red/Blue:** A modified version where one player is red and the other is blue, potentially influencing object perception.

## H.4   Breakout Variants

**Breakout:** The original *Breakout* environment where the player controls a paddle to break bricks.

**All Blocks Red:** All blocks are red, removing the color of blocks that do not hold game-relevant information.

**Player/Ball Red:** The paddle is slightly redder than usual, potentially changing agent perception and behavior.

## H.5   Freeway Variants

**Freeway:** The standard *Freeway* environment where a chicken crosses a highway with moving cars.

**All Cars Black:** All vehicles are black, reducing visual diversity between the cars. There are no additional changes.

**Stop All Cars:** All cars are stopped on the edge of the frame, making it trivial to pass the street.

## H.6   Frostbite Variant

**Frostbite:** The original *Frostbite* environment where the player builds an igloo while avoiding hazards. To collect ice, the player has to jump between moving ice shelves.

---

[4]https://ale.farama.org/

**Static Ice:** Ice platforms remain fixed instead of moving, altering difficulty (making it much easier).

### H.7 MsPacman Variants

**MsPacman** The standard *MsPacman* environment that is very similar to the Pac-Man environment.

**2nd Level:** A later level with a changed maze structure. No agent ever reached this level in training. As such, it is out of distribution for all agents.

### H.8 Pong Variants

**Pong:** The standard *Pong* environment where two paddles hit a ball back and forth.

**Lazy Opponent:** The opponent stays still while the ball is flying away from it and only starts moving after the player hits the ball. No visual changes are visible. This was presented as one of two examples of misalignment in Delfosse et al. (2024c).

**Hidden Opponent:** The opponent is hidden from the player. The only observable objects are the paddle and the Ball. This follows the experiment by Delfosse et al. (2024c) and is used to directly compare with their work.

### H.9 Riverraid Variants

**Riverraid:** In *Riverraid*, you control a jet that flies over a river: you can move it sideways and fire missiles to destroy enemy objects. Each time an enemy object is destroyed, you score points (i.e., rewards).

**Color Change:** Change the color of all objects to a different preset color—no change in the game logic.

**Linear River:** Fix the river to always have the same shape—a single line—and width: no splits, etc.

### H.10 SpaceInvaders Variants

**SpaceInvaders:** In *SpaceInvaders*, your objective is to destroy the space invaders by shooting your laser cannon at them before they reach the Earth. The game ends when all your lives are lost after taking enemy fire or when they reach the earth.

**Shields off by 3:** The shields are moved 3 pixels to the right. No further changes.

Table 15: The HackAtari modifications for all game variants used in our work.

| | |
|---|---|
| **Amidar** | |
| Player to Roller | `paint_roller_player` |
| Enemy to Pig | `pig_enemies` |
| **Bowling** | |
| Shift Player | `shift_player` |
| **Boxing** | |
| Boxers Red/Blue | `color_player_red, color_enemy_blue` |
| **Breakout** | |
| All Blocks Red | `color_all_blocks_red` |
| Player/Ball Red | `color_player_and_ball_red` |
| **Freeway** | |
| All Cars Black | `all_black_cars` |
| Stop All Cars | `stop_all_cars_edge` |
| **Frostbite** | |
| Static Ice | `reposition_floes_easy` |
| **MsPacman** | |
| Level 2 | `set_level_1` |
| **Pong** | |
| Lazy Opponent | `lazy_enemy` |
| Hidden Opponent | `hidden_enemy` |
| **Riverraid** | |
| Color Change | `game_color_change01` |
| Linear River | `linear_river` |
| **SpaceInvaders** | |
| Shields off by 3 | `relocate_shields_off_by_three` |

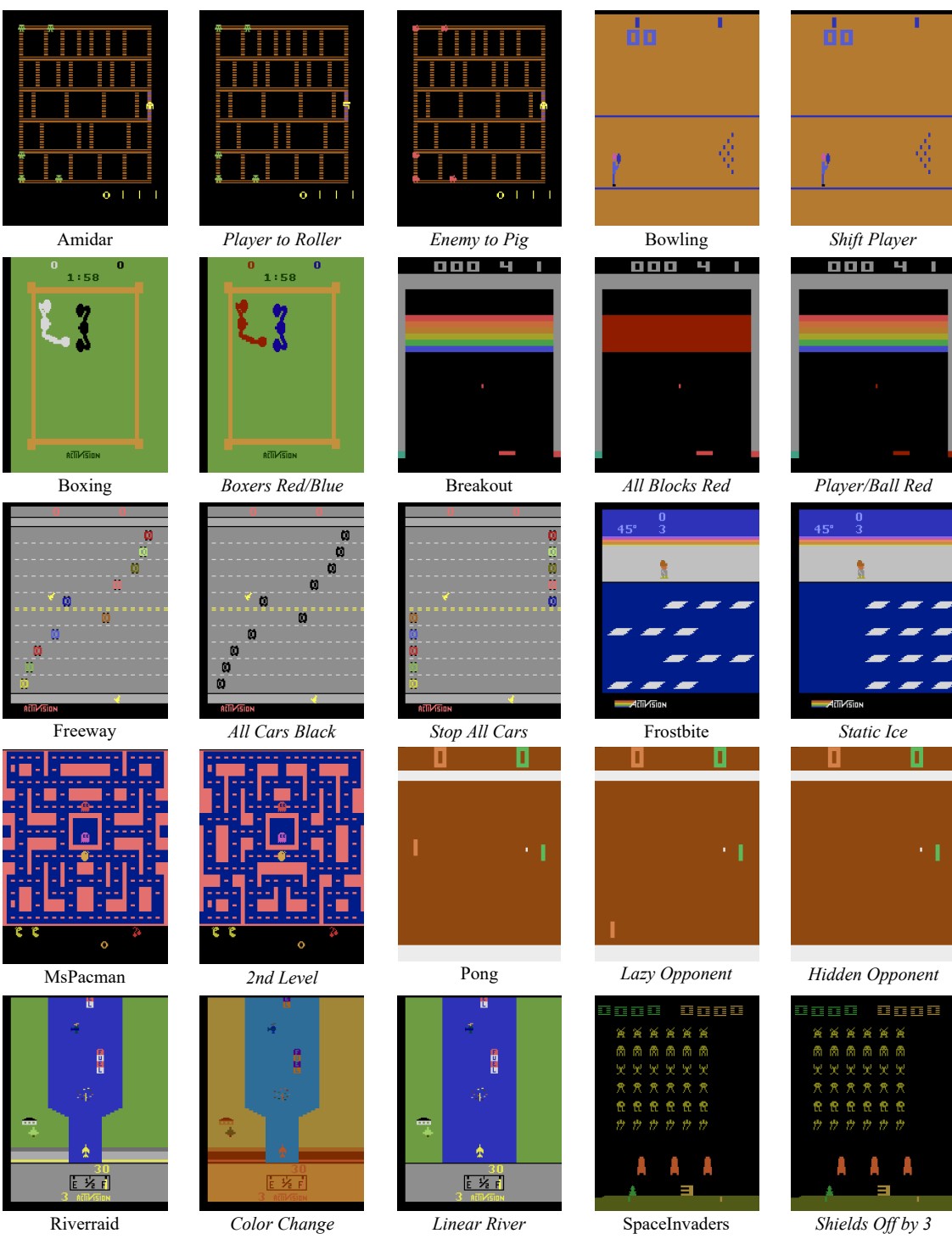

Figure 14: Illustration of different game variants used in our study. Games are grouped, starting with the original game on the left, followed by the variants (*italic* label). Groups may break to the next row. These variants introduce modifications, such as color changes and variations in (enemy) behavior, to study their impact on learning and generalization.

