# OpenReview forum: "Do Object Channels Improve Robustness in Deep Reinforcement Learning?"
_TMLR — Accepted by TMLR_

### Review · Reviewer_92JT · 2026-04-13

**Summary Of Contributions:**

This paper investigated whether object channels can improve the robustness of deep reinforcement learning, especially visual reinforcement learning. It proposes spatially grounded Object&Pixel-Channels (OPC), which serve as an inductive bias for deep RL, isolating representation geometry from architectural effects. The extensive experiments across Atari environments show that OPC improves zero-shot robustness to perturbations.

Strength: The work presents a clear and well-controlled evaluation, and covers multiple experimental setups and perturbation types.

Weakness: The paper lacks a theoretical analysis to explain the reason for the improvement of robustness in deep RL.

**Audience:**

Yes

**Audience Explanation:**

This work gives a good understanding of how input representation affects robustness in deep RL. It highlights the role of spatially grounded object-centric features in mitigating shortcut learning and provides useful empirical evidence to show how it improves zero-shot robustness under visual perturbations.

**Claims And Evidence:**

Yes

**Claims Explanation:**

This paper conducted comprehensive experiments to evaluate the effectiveness of OPC and used clear metrics and controlled perturbations to make it a strong empirical evaluation.

**Requested Changes:**

1. The paper shows the efficiency of OPC and PPO under fixed wall-clock time constraints in Figure 6(a). Can authors also provide the comparison of GPU memory usage (or peak memory footprint)?

2. Can authors provide a clear mathematical objective being optimized in this paper and clarify how the proposed representation connects to improving robustness?

3. In section 3.4, the paper mentions “Object channels increase input dimensionality proportionally to the number of object categories $|C|$”. What will happen if object categories are merged and not clearly split?

---

> ### Author Response · Authors · 2026-04-21
>
> We thank you for the positive assessment and the three constructive requests. We address each in turn and describe the corresponding additions to the paper. We will update the paper draft in the next few days, after also addressing the points raised by the other reviewers.
>
> ---
>
> **RC1: GPU Memory Usage**
>
> We will report GPU memory statistics logged throughout training via `torch.cuda.memory_allocated()` and `torch.cuda.memory_reserved()`. Results show for PPO, peak reserved memory increases from 0.5 GB to 3.3 GB in Frostbite (the largest representation with 12 channels), scaling roughly linearly with the number of channels as expected. For Rainbow, the larger replay buffer makes the memory impact more pronounced, which is why we reduced its size from $10^6$ to $10^5$ transitions for OPC experiments (Appendix D.3). We have added a table reporting peak reserved and allocated memory per game to the paper (Appendix F.3).
>
> ---
>
> **RC2: Mathematical Objective and Connection to Robustness**
>
> Our paper intentionally uses the same RL objectives as the pixel-only baselines. The contribution is at the representation level, not the training objective. The connection between OPC and robustness can be stated precisely: under a HackAtari perturbation that alters the visual rendering of a frame without changing object positions or game dynamics, the object-channel component $C_t$ is exactly invariant by construction (Eq. 2), since the bounding boxes of objects are unchanged. This means $X_t$ can change only through the pixel channel $D_t$. In terms of state abstraction theory, the mapping $I_t &rarr; C_t$ defines a lossless abstraction with respect to object identity and position, collapsing the equivalence class of all frames sharing the same object layout. This is precisely what reduces or even prevents shortcut learning: spurious pixel-level features within the same equivalence class are suppressed in $C_t$. They therefore cannot be exploited by the convolutional backbone.
>
> In our work, we further investigated whether the suppression in $C_t$ is sufficient, since pixel-level features are still accessible in $X_t$, and we evaluated additional perturbations that manipulated enemy behavior or object positions to create out-of-distribution situations. We have added a concise, formal proposition in Section 2 that captures this theoretical idea and clarifies the connection between representation and robustness.
>
> ---
>
> **RC3: Merged Object Categories**
>
> We conducted a small preliminary ablation exploring category merging, comparing our standard per-type channel representation with a semantically grouped variant that merges object categories sharing similar behavioral roles into a single channel, e.g., all collectible objects into a single channel.
> While this, of course, reduces the problem of exploding channel numbers, the results were mixed, game-dependent, and highly specific to the merged set of channels: in environments with sparse or role-sharing objects, merging proved beneficial or neutral (compared to OPC), while in environments where fine-grained category distinction was needed, performance degraded. Crucially, no consistent trend emerged across games, and the direction of the effect could not be predicted from game properties alone. We omit this analysis from the present paper for two reasons. First, the inconclusive results on the effect of merging depend on per-game object dynamics, which require a dedicated, larger-scale study. Second, our primary goal is to isolate the effect of representation within a controlled, fixed ontology; varying the ontology would broaden this analysis and dilute it from this primary goal. We agree that this is an important direction and now list it as future work in Section 4.

---

### Review · Reviewer_Uaxc · 2026-04-17

**Summary Of Contributions:**

This paper proposes OPC and investigates whether object-channel representations improve robustness in deep reinforcement learning. The work examines the abstraction-fidelity trade-off, sensitivity to detector noise, and scalability of such representations across multiple Atari environments under visual perturbations.

**Audience:**

No

**Audience Explanation:**

1. The paper lacks discussion and comparison with OCCAM (Deep Reinforcement Learning via Object-Centric Attention), which shares highly similar motivations, methods, and evaluation focus (robustness). If OCCAM has already done substantially the same thing, the novelty and contribution of this paper need to be re-evaluated.

2.  Incomplete reporting of detector quality in Experiment 3. The paper uses YOLO and RT-DETR but does not report standard detection metrics such as mAP or F1 scores in the tables. Only downstream policy rewards are shown. Providing these metrics would help readers disentangle whether performance drops are due to detector quality or policy sensitivity.

3. Table 1 reports OPC results using an oracle detector (OCAtari REM). While this helps isolate the effect of the representation, it introduces additional information (perfect object locations) that pixel baselines do not receive. At a minimum, the paper should include a column with OPC using a real detector (e.g., YOLO).

4. Misattribution of failure. Riverraid failure is blamed on "over-abstraction," but the real issue is the detector simply doesn't extract riverbanks. This is a detector ontology problem, not a representation flaw.

**Claims And Evidence:**

Yes

**Claims Explanation:**

1. Clean attribution. The paper fixes the algorithm and network architecture, varying only the input representation. This cleanly isolates the effect of the representation itself.

2. Clear robustness demonstration. In discriminative environments like Pong, the paper clearly shows OPC maintains stability while pure object channels collapse.

**Requested Changes:**

See above

---

> ### Author Response · Authors · 2026-04-24
>
> We would like to thank you for your time and for the questions raised. We have carefully considered all comments and revised the manuscript to further emphasize key details and improve the clarity of our presentation.
>
> ---
>
> **RC1: Comparison with OCCAM**
>
> The reviewer notes a strong overlap with an arXiv preprint.
>
> We agree that it presents a similar methodology/framework. However, the current submission extends those preliminary findings by the following points:
>
> First, scope. OCCAM treats the Planes ($C_t$) as one of four masking strategies and evaluates them at a high level across only six environments. Our paper takes OPC (Planes + DQN-like) as its central object of study. It provides a **substantially deeper analysis**: Systematic perturbation evaluation via HackAtari, a formal definition of $C_t$ (the Planes), detector noise ablation with YOLO and RT-DETR, and GPU memory and wall-clock scaling analysis. None of which appear in the referenced paper, OCCAM.
>
> Second, the **hybrid representation**. Our paper introduces and studies OPC (Object & Pixel Channels), which combines object information (object channels) with the raw pixel observation. OCCAM does not investigate this combination to get the best of both worlds.
>
> Third, **formal grounding**. OCCAM is empirical throughout. This paper adds a formal proposition connecting the representation geometry to robustness guarantees under perturbation, and situates the contribution within state abstraction theory.
>
> We are happy to update the Related Work section to state this arXiv preprint.
>
> ---
>
> **RC2: Incomplete Reporting of Detector Quality**
>
> The requested detection metrics are reported in the paper's appendix, though they may not be referenced prominently enough. In the main text, Figure 5b shows per-class F1 scores for both YOLO and RT-DETR in Pong, and Appendix F.4 provides mAP@50 and mAP@50--95 curves for Freeway, Amidar, and Pong as well as F1 scores. The key point, that detection quality (mAP) is needed for policy success, is stated explicitly in Section 3.3. We agree that mAP and F1 are important metrics for measuring detection quality.
>
> To improve accessibility to these details, we have cross-referenced Appendix F.4 directly in the caption of Figure 5.
>
> ---
>
> **RC3: Oracle Detector in Table 1**
>
> As correctly pointed out by the reviewer, using the oracle detector results in Table 1 helps isolate the effect of representation. That’s why we report these values instead of those from real detectors.
>
> On the other hand, we politely disagree with the reviewer that OPC adds information. OPC ($X_t$) concatenates $C_t$ to $D_t$. $C_t$ consists of different channels with bounding boxes of objects present in $D_t$. Therefore, $C_t$ is an abstraction of $D_t$ and cannot include more information than $C_t$, and neither does $X_t$.
>
> We agree, however, that results on imperfect detectors are also worth considering. For this reason, we report them in Figure 5a and Figure 11 (Appendix).
>
> Including these values in Table 1 is a matter of personal preference. We acknowledge the reviewer's opinion, but we will stick to our choice, as it provides a fairer comparison with SemVec and SCoBots, which use the same oracle detector, and with $D_t$, which has access to the same information as $X_t$ (see above).

---

> ### Author Response · Authors · 2026-04-24
>
> **RC4: Misattribution of Failure**
>
> We agree that detection precedes abstraction: one can only choose to omit what can first be detected, and with our current detector, riverbanks cannot be extracted, making it difficult to cleanly separate an ontological design choice from a detector limitation.
>
> With the term "over-abstraction," we highlight a broader design question that we believe is fundamental to object-centric RL: *What is an object?* Even if detection was feasible, riverbanks could conventionally be excluded from an object-centric ontology, as they are a global spatial background structure rather than a discrete, localized, independently moving entity. Nevertheless, they are clearly task-relevant, and without them, the agent loses the ability to navigate.
>
> This raises a genuinely philosophical question: Is task relevance alone sufficient to qualify something as an object? If yes, the boundary between "object" and "background" becomes blurred. This further highlights the benefit of a hybrid representation like OPC, with the pixel representation as a backup if the ontology is drawn too narrowly.
>
> We also note that Reviewer Ryf3 explicitly highlighted the abstraction--fidelity trade-off as "a fundamental question in representation learning that resonates beyond reinforcement learning," which further supports the broader significance of this discussion.
>
> To address this, we updated Section 3.2 to: (1) acknowledge the reviewer's point that the detector limitation and the ontological design choice are empirically difficult to disentangle with the current setup; (2) explicitly pose the question of what qualifies as an object; and (3) clarify that "over-abstraction" refers to the broader consequence of discarding task-relevant information, independent of its cause.

---

### Review · Reviewer_Ryf3 · 2026-04-19

**Summary Of Contributions:**

This paper systematically investigates the role of spatially grounded semantic channel representations, termed Object&Pixel-Channels (OPC), in deep reinforcement learning. The main contributions are as follows:
1. In a multi-environment setting, OPC is shown to substantially improve zero-shot robustness to visual perturbations compared to pure pixel inputs, while maintaining or even improving in-distribution performance.
2. The work clearly reveals an abstraction-fidelity trade-off in object-centric representations: pure object channels (without pixels) can suffer catastrophic failure when the object ontology is incomplete (e.g., missing background geometric information), whereas the hybrid OPC representation effectively mitigates this issue by retaining the pixel channel as a "visual safety net."
3. The relationship between OPC's computational overhead and sample efficiency is analyzed, demonstrating that the sample efficiency gains conferred by the semantic inductive bias often offset the reduced throughput caused by increased input dimensionality.

**Audience:**

Yes

**Audience Explanation:**

This paper attracts its intended audience for:
1. Rather than proposing a novel and complex algorithm, it provides an in-depth analysis of how a specific representation design choice influences downstream policy behavior.
2. It offers a systematic evaluation framework along with reproducible experimental procedures, which will be of practical value to researchers aiming to improve the reliability of deep reinforcement learning.
3. The examination of the abstraction-fidelity trade-off touches upon a fundamental question in representation learning -- what information to preserve versus discard. This theme that resonates beyond reinforcement learning and carries relevance for the broader machine learning community.

**Broader Impact Concerns:**

The paper does not involve personal data, privacy issues, or techniques that could be readily misused for generating bias or
adversarial attacks.

**Claims And Evidence:**

Yes

**Claims Explanation:**

The majority of the core claims are well supported by the evidence. Robustness improvement:
1. The data presented in Figure 4 and Table 1 clearly demonstrate the generalization advantage of OPC under visual perturbations from HackAtari (e.g., Hidden Opponent Pong, color alterations in Boxing), whereas pure pixel-based baselines suffer from policy collapse.
2. Abstraction-fidelity trade-off: The case study on Riverraid (Table 2) provides a compelling demonstration that pure object channels, which lack background geometric information, lead to severe state aliasing and a dramatic drop in performance, while the hybrid OPC representation ($\mathbf{X}_t$) successfully overcomes this limitation.
3. Sample efficiency: The convergence curves shown in Figure 6 indicate that OPC frequently reaches higher performance levels more quickly, supporting the claim that the semantic inductive bias improves sample efficiency.

**Requested Changes:**

1. Include direct comparisons with data augmentation methods.
The current evaluation does not allow the reader to assess the incremental benefit of OPC relative to mainstream techniques for improving visual robustness in RL, such as RAD or DrQ. Add PPO + RAD (or DrQ) as an additional baseline in at least three or four core environments (e.g., Pong, Breakout, Amidar, Riverraid). Analyze whether the robustness gains of OPC under HackAtari visual perturbations are complementary to or overlapping with those conferred by data augmentation. This is essential for positioning the practical value of OPC.

2. Extend and strengthen the experiments with learned object detectors.
The current learned-detector experiments are limited to YOLO and RT-DETR and show poor performance in several settings, which does not substantiate the claimed advantage of modular compatibility with state-of-the-art perception models. (1) Integrate a stronger, off-the-shelf detector, such as Grounding DINO (using text prompts) or SAM (with class prompts), to demonstrate how the OPC framework benefits from improved perception. (2) If training or deploying a stronger detector is infeasible, the paper should more explicitly discuss the limitations of the current detectors and tone down the claim that detection models can be easily upgraded without retraining the policy. This should instead be framed as a hypothesis to be tested or as future work.

3. Explicitly delineate the limitations concerning structural generalization. In Section 4 (Discussion) or Section 3.2, add a paragraph that explicitly discusses the boundary conditions under which OPC fails, using the MsPacman maze-layout change as a concrete example. Clarifying that visual robustness does not imply structural generalization will help readers better understand the method's scope of applicability.

4. The ablation study is not sufficiently thorough. The authors compare OPC with pure object channels, but they do not include control groups such as "pixel channels plus random masks" or "pixel channels plus additional convolutional kernels." Such ablations would help rule out the alternative explanation that the mere addition of extra input channels—rather than the semantic content of those channels—accounts for any observed gains in capacity or performance.

5. Correct minor typographical and presentation issues. (1)On page 6, "re- and grayscaled" is missing "sized".(2)When mentioning the reduction of Rainbow's replay buffer size (Appendix D.3), briefly note the potential impact on baseline performance to ensure a fair interpretation of the comparisons.

---

> ### Author Response · Authors · 2026-04-23
>
> We thank you for your thoughtful and systematic evaluation of our work.
> Below, we detail how we have addressed each requested change in the revised manuscript, which we will upload when we have also addressed the other reviews.
>
> ---
> **RC 1: Comparison with data augmentation methods**
>
> We acknowledge that RAD and DrQ are important baselines for robustness. However, we wish to clarify a fundamental conceptual distinction: while data augmentation methods like RAD improve robustness by *diversifying the training distribution*—forcing the agent to learn invariance to pixel-level transformations—OPC improves robustness through *inductive biases in the representation*.
>
> As demonstrated in the newly added Proposition 1, the object channel component $C_t$ is inherently invariant to visual perturbations that preserve task structure and object characteristics. This invariance is a structural property of OPC, independent of the augmentations encountered during training. Because these mechanisms are fundamentally different, we view them as complementary rather than competing.
>
> We agree with the reviewer that an empirical comparison would provide valuable insights for practitioners. Consequently, we will include PPO+RAD as a baseline for Pong, Breakout, Amidar, and Riverraid to analyze whether these robustness gains are additive or overlapping. We note, however, that standard RAD augmentations (e.g., random crop, cutout) do not align with the specific distribution shifts in HackAtari, which may limit the effectiveness of pure augmentation in this particular evaluation protocol.
>
> Finally, future work should further explore the synergy between domain randomization and image augmentation (e.g., RAD) for OOD generalization in HackAtari. Specifically, training agents across a suite of HackAtari visual perturbations would enable a rigorous assessment of how diversifying the training affects robustness to held-out distribution shifts.
> However, as these techniques represent a distinct orthogonal approach to robustness via data distribution rather than representation, a full investigation of their intersection is out of scope for this work and a promising direction for complementary future research.
>
> The short ablation will be added in Appendix F.7, showing the difference in robustness to the HackAtari modifications between PPO+RAD and OPC.
>
> Our results indicate that while RAD can enhance generalization, it is less consistent than OPC, providing gains in Riverraid but failing to handle the color-perturbed variant. This discrepancy highlights the aforementioned: pixel-level augmentations do not necessarily capture the structural invariances inherent to our representational approach. Furthermore, we observed that excessive augmentation can be counterproductive, leading to significant performance stagnation.
>
> ---
> **RC2: Extend and strengthen the experiments with learned object detectors**
>
> In the last days, we investigated both Grounding DINO and SAM3 as stronger off-the-shelf alternatives, prompted with object class names for each game. Unfortunately, neither model produces usable detections on Atari frames out of the box — the domain gap between natural images and Atari sprites is substantial enough that both models require a similar fine-tuning procedure we applied to YOLO and RT-DETR (cf. Appendix D.4).
>
> We therefore agree with the reviewer's second suggestion and will tone down the claim in Section 4. Whether a stronger detector further improves downstream performance is, in itself, an open question that we will explicitly frame as a hypothesis and future work.
>
> ---
> **RC3: Limitations**
>
> We will extend our discussion of the limitations and go into more detail about the failure modes, especially regarding maze and/or map layouts, and highlight that visual robustness does not imply generalization to structural changes.
>
> Section 4 already includes a dedicated paragraph ("From Visual to Structural Generalization") that discusses the MsPacman maze-layout failure and the boundary between visual robustness and structural generalization. The Conclusion also repeats this point explicitly. We agree, however, that this limitation deserves more prominent treatment earlier in the paper. We will add a concise paragraph in Section 3.2 after the Riverraid analysis, using MsPacman as a concrete example and clearly stating that OPC's invariance guarantees (new Proposition 1) apply only to rendering-level perturbations and do not extend to structural changes in environment topology.

---

> ### Author Response · Authors · 2026-04-23
>
> **RC4: Ablation study**
> Did we understand you correctly? The key question here is whether OPC's performance gains stem from the semantic content of the object channels or simply from the addition of extra input channels?
>
> If this is the question, we propose an alternative to testing it: replace the semantic object channels with duplicates of the pixel channel, increasing the input dimensionality and, as a result, the parameter size identical to OPC while removing all semantic structure. Injecting random noise could undermine training stability, making it difficult to cleanly attribute performance differences to the presence or absence of semantic content. We tested the duplication idea across 4 environments. The results showed no particular difference in performance and robustness compared to the simple baseline PPO agent. Runtime was, as expected, similar to that of OPC. The performance metrics are in Appendix F.6.
>
> ---
> **RC5: Minor Issues**
>
> Thank you for pointing these out. We fixed them.

---

### Author Response · Authors · 2026-04-24
**Update Manuscript**

We have uploaded a revised version of our manuscript, which incorporates several improvements to the theoretical grounding, experimental analysis, and overall clarity.

All significant changes in the text are marked with [new] or [fix] for easy navigation. The key updates include:
* We now cite Blüml et al. (2025) [1] in Sections 1 and 5. We acknowledge their influential work on how inductive biases improve robustness and provide a concise comparison to our approach.
* We introduced Proposition 1 (Section 2.1) to formally characterize the relationship between our input representation and theoretical robustness to environmental visual shifts.
* Updated Section 3.2 to specifically address Reviewer Uaxc’s concerns regarding detection failures and ontological design choices.
* We have toned down the claim that stronger object detectors inherently yield better results (Section 4); this is now framed as a hypothesis for future empirical validation.
* Added a short extension to Section 4 regarding ontology granularity (e.g., merging channels) and the broader question of optimal representation.
* Added a GPU memory utilization table in App. F.3.
* Added App. F.6: New results investigating whether performance gains stem from increased input size or underlying semantic information.
* Added App. F.7: A comparative analysis of whether RAD offers robustness improvements similar to OPC.


[1] Jannis Blüml, Cedric Derstroff, Bjarne Gregori, Elisabeth Dillies, Quentin Delfosse, and Kristian Kersting. Deep reinforcement learning via object-centric attention. arXiv preprint arXiv:2504.03024, 2025.

---

### Decision · Action_Editor_Y7bQ · 2026-06-04

**Recommendation:** Accept as is

**Additional Comments:**

This submission underwent a productive revision process where the authors systematically addressed the concerns of all three reviewers. All three reviewers moved to 'Leaning Accept' after the revisions, indicating satisfaction with the technical rigor. The paper's primary strength lies in its disciplined experimental design that cleanly attributes performance differences to the input representation rather than confounding factors. The hybrid OPC design is well-motivated by the demonstrated failure modes of both pure pixel and pure object-channel approaches, and the abstraction-fidelity trade-off framing provides a useful conceptual contribution beyond mere empirical results. The remaining concerns are minor and do not materially affect the technical soundness, supporting acceptance.

**Audience:**

Yes

**Audience Explanation:**

This paper will interest researchers working on representation learning for deep reinforcement learning, particularly those focused on object-centric methods, visual robustness, and shortcut learning. The systematic evaluation framework and reproducible experimental procedures offer practical value for improving RL reliability. The exploration of the abstraction-fidelity trade-off connects to fundamental questions in representation learning about what information to preserve versus discard, a theme that resonates beyond RL into the broader machine learning community. Researchers studying distribution shift, sample efficiency, and hybrid representations in pixel-based RL will find this work informative.

**Claims And Evidence:**

Yes

**Claims Explanation:**

The technical claims are well-supported by experimental evidence across multiple Atari environments. The robustness improvements are convincingly demonstrated in Figure 4 and Table 1, where OPC maintains performance under HackAtari perturbations (e.g., Hidden Opponent Pong, color alterations in Boxing) while pure pixel-based baselines suffer policy collapse. The abstraction-fidelity trade-off is compellingly shown through the Riverraid case study (Table 2), where pure object channels fail without background geometric information while the hybrid OPC succeeds. The authors effectively addressed reviewer concerns by adding PPO+RAD baselines (Appendix F.7), a duplicate pixel channel ablation (Appendix F.6) to rule out capacity-based explanations, GPU memory analysis (Appendix F.3), and Proposition 1 for formal grounding. The Riverraid failure misattribution was resolved by disentangling detector limitations from ontological design choices. The methodology is rigorous: the algorithm and architecture are fixed, varying only the input representation, which cleanly isolates the effect of representation on policy behavior.